# Leveraging Predictive Equivalence in Decision Trees

Hayden McTavish [* 1]  Zachery Boner [* 1]  Jon Donnelly [* 1]  Margo Seltzer [2]  Cynthia Rudin [1]

## Abstract

Decision trees are widely used for interpretable machine learning due to their clearly structured reasoning process. However, this structure belies a challenge we refer to as predictive equivalence: a given tree's decision boundary can be represented by many different decision trees. The presence of models with identical decision boundaries but different evaluation processes makes model selection challenging. The models will have different variable importance and behave differently in the presence of missing values, but most optimization procedures will arbitrarily choose one such model to return. We present a boolean logical representation of decision trees that does not exhibit predictive equivalence and is faithful to the underlying decision boundary. We apply our representation to several downstream machine learning tasks. Using our representation, we show that decision trees are surprisingly robust to test-time missingness of feature values; we address predictive equivalence's impact on quantifying variable importance; and we present an algorithm to optimize the cost of reaching predictions.

## 1. Introduction

Decision trees are widely used for interpretable machine learning (Rudin et al., 2022). Their structure of discrete decisions has long been leveraged for difficult tasks such as handling missing data (Therneau et al., 1997) and measuring variable importance (Breiman, 1984). Recent advances in decision tree optimization (Lin et al., 2020; Demirović

*Equal contribution  [1]Department of Computer Science, Duke University, Durham, North Carolina, USA  [2]Department of Computer Science, University of British Columbia, Vancouver, BC, Canada. Correspondence to: Hayden McTavish <hayden.mctavish@duke.edu>, Zachery Boner <zachery.boner@duke.edu>.

*Proceedings of the 42$^{nd}$ International Conference on Machine Learning*, Vancouver, Canada. PMLR 267, 2025. Copyright 2025 by the author(s).

Code for our algorithms and experiments can be found at https://github.com/HaydenMcT/predictive-equivalence

et al., 2022; Aglin et al., 2020) – including algorithms for enumerating the entire set of near-optimal decision trees (the Rashomon set; Xin et al., 2022) – have garnered substantial research interest. These advances have enabled new perspectives on predictive multiplicity (Marx et al., 2020; Watson-Daniels et al., 2023) and variable importance (Dong & Rudin, 2020; Fisher et al., 2019; Donnelly et al., 2023).

However, decision trees can be misleading, because they correspond not just to a classifier but also to a *particular way of evaluating the classifier*. Consider the two equivalent trees in Figure 1. The two trees encode the same logical AND decision function, but they suggest different orders of querying $X_1$ and $X_2$. A practitioner would typically deploy only one of these trees, but either order is equally justified.

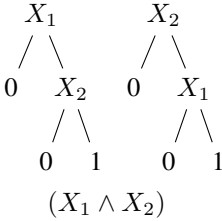

*Figure 1.* Two decision trees, suggesting a different evaluation order, but which represent the same logical formula $(X_1 \land X_2)$.

This phenomenon, which we call *predictive equivalence* (Sober, 1996), poses several distinct challenges:

**(1)** Decision trees imply an evaluation procedure that can get stuck on irrelevant missing information. If $x_1$ is missing and $x_2 = 0$, the first tree in Figure 1 cannot be traversed, but the second tree clearly predicts 0.

**(2)** Tree-based variable importance metrics change across predictively equivalent trees. For example, Gini importance will suggest that $x_2$ is more important to the first tree in Figure 1, even though this order is arbitrary.

**(3)** Logically equivalent trees with different evaluation orders appear in the Rashomon set as distinct trees. This phenomenon causes some models to be over-represented in the Rashomon set, which biases some downstream tasks.

**(4)** A decision tree implies a constrained order for evaluating variables, but this order may be sub-optimal when each variable has an associated cost.

We provide a representation of decision-tree classifiers that abstracts away the evaluation order. To do this, we convert decision trees into disjunctive normal form (DNF; an OR of ANDs) and reduce to a minimal set of sufficient conditions for making predictions. This representation allows us to address the above challenges: we uncover many cases where decision trees can still make predictions despite some variables being missing, we make variable importance metrics for trees more reliable, we resolve predictive equivalence in the Rashomon set, and we optimize the cost of variable acquisition needed to reach a prediction using a tree.

## 2. Related Work

### 2.1. Decision Trees and Simplicity

There is a substantial body of work on decision tree learning. Greedy decision tree algorithms, such as CART (Breiman, 1984) and C5.0 (Quinlan, 2014), find decision trees in a greedy top-down, recursive manner. The GOSDT algorithm by Lin et al. (2020), the DL8.5 algorithm by Aglin et al. (2020), and the MurTree algorithm by Demirović et al. (2022) provide methods to optimize the decision tree hypothesis space directly. A range of other approaches also afford optimal decision trees via more general solvers (Bertsimas & Dunn, 2017; Verwer & Zhang, 2019). These algorithms can be used to find highly accurate decision trees – indeed, well-optimized single decision trees can approach the performance of decision tree ensembles (Vidal & Schiffer, 2020; McTavish et al., 2022), which are often state of the art for tabular data (Grinsztajn et al., 2022). Our representation of decision trees applies to trees discovered via any method.

Our work is particularly related to the problem of explanation redundancy in decision trees. This concept is explored by Izza et al. (2022), who demonstrate that the paths taken through the tree to reach predictions ("path explanations") often have redundant variables in them, which are not necessary to make the prediction. The authors present polynomial-time algorithms to compute succinct path explanations. In contrast, we present a method to compute a minimal boolean logical representation of the entire decision tree, using the Quine-McCluskey algorithm (Quine, 1952; McCluskey, 1956) as a subroutine. This representation enables succinct path explanations of predictions for free, once the global representation is computed for some up-front cost. Our representation also enables several downstream applications beyond prediction explanations.

A line of work on the simplicity of machine learning models shows that when data has noise in the outcomes (common on many tasks we consider in our experiments), simpler decision trees will be competitive in performance with more complicated ones (Semenova et al., 2022; 2023; Boner et al., 2024). If our decision trees have a small number of leaves,

the number of variables in the Quine-McCluskey subroutine will be small, and our algorithm for simplification will be efficient despite the NP-completeness of the problem.

### 2.2. Applications

**Variable Importance.** Decision trees have been used for variable importance since at least the introduction of random forests (Breiman, 2001a). Notably, specialized metrics that quantify importance based on the reduction in impurity achieved when splitting on a particular feature have been developed to measure variable importance in decision trees (Louppe et al., 2013; Kazemitabar et al., 2017). In Section 5.1, we show that predictively identical trees can yield very different impurity reduction values.

There are also metrics such as SHAP (Lundberg & Lee, 2017), permutation importance (Breiman, 2001a; Fisher et al., 2019), conditional model reliance (Fisher et al., 2019), LOCO (Lei et al., 2018), and LIME (Ribeiro et al., 2016), that quantify variable importance based on permuting data across a particular decision boundary. These metrics are invariant to predictive equivalence, because they evaluate only the decision boundary.

Recent work examines variable importance over all models in the set of near-optimal models (Fisher et al., 2019; Dong & Rudin, 2020; Donnelly et al., 2023), i.e., the Rashomon set (Breiman, 2001b; Rudin et al., 2024), rather than a single model. Of particular note, the Rashomon Importance Distribution (RID) (Donnelly et al., 2023) demonstrated that the stability of variable importance estimates can be improved by examining the distribution of variable importances over Rashomon sets computed on bootstrapped datasets. In Section 5.2, we show that predictive equivalence within each Rashomon set confounds the practical implementation of RID, and we show how to correct this.

**Missing Data.** A popular approach for dealing with missing feature values is to impute them – with either a simple estimator such as the mean, or a function of the other covariates. For background on imputation, see Shadbahr et al. (2023); Emmanuel et al. (2021); Van Buuren & Oudshoorn (1999). Multiple imputation accounts for uncertainty in imputation by combining results from several estimates (Rubin, 1988; Van Buuren & Oudshoorn, 1999; Schafer & Graham, 2002; Stekhoven & Bühlmann, 2012; Mattei & Frellsen, 2019). There is also a body of work regarding surrogate splits, a tree-specific approach which learns alternative splits to make when a variable is missing (Therneau et al., 1997; Breiman, 1984). Each of these approaches introduces bias when the probability of a variable being missing depends on the variable's underlying value, beyond what can be modeled by the covariates – this setting is referred to as Missing Not at Random (MNAR) (Little & Rubin, 2019). We show

that our proposed representation reveals examples whose predictions are identical under any form of imputation.

Imputation can be detrimental to prediction when missingness provides information about the label. There are a wide range of theoretical and empirical findings supporting the need to reason explicitly on missingness in this setting (Sperrin et al., 2020; Le Morvan et al., 2021; Van Ness et al., 2023; Stempfle et al., 2023; McTavish et al., 2024). Many such approaches are tree or tree-ensemble specific, leveraging the simple structure of trees (Kapelner & Bleich, 2015; Twala et al., 2008; Beaulac & Rosenthal, 2020; Therneau et al., 1997; Wang & Feng, 2010; Chen & Guestrin, 2016). However, such missingness-specific modeling requires sufficient observation of missingness at training time. When a missingness pattern occurs only at test time, or when the missingness mechanism has a distribution shift from training time (the latter setting being particularly common in medical domains, e.g., Groenwold, 2020; Sperrin et al., 2020), it is difficult to learn missingness-specific patterns.

Stempfle & Johansson (2024) propose a metric to measure how often models rely on features with missing values, and they propose a model class designed to be robust to missingness. Their scoring model MINTY uses logical disjunctions such that, if any term in a disjunction is known to be true, that entire disjunction can be evaluated. This allows one variable to serve as a backup when another is missing. We show that our representation for decision trees dramatically improves the trees' performance on this metric, without changing the decision boundary.

**Cost Optimization.** Many real-world problems have a cost to acquire variables – for example, ordering an MRI is expensive and time-consuming. Many types of costs have been studied (Turney, 2002), but our focus is on test cost, or the minimum cost associated with obtaining a prediction from a model. There are many cost-sensitive decision tree algorithms in the literature (Lomax & Vadera, 2013; Costa & Pedreira, 2023). However, all of these approaches directly optimize a decision tree to account for these costs and use that tree top-down at test time, even if there is a more cost-effective way to obtain predictions from the same tree. In contrast, we optimize the cost of evaluating predictions from *a given decision tree* (cost-optimal or otherwise) when some or all variables are unknown. Note that we assume each feature has a fixed cost across all samples.

We optimize the cost of applying a decision tree by applying *Q-learning* (Watkins, 1989). Q-learning is a model-free approach for policy learning that estimates the value of each action in each state by allowing an agent to explore the state space. During exploration, the reward of the $j$-th visited state is gradually propagated back to the $(j-1)$-th state. Given sufficient *episodes* – iterations of this exploration –

it has been shown that Q-learning will produce the optimal policy for a given problem (Watkins, 1989; Watkins & Dayan, 1992). Since the introduction of Q-learning, the field of reinforcement learning has dramatically expanded. We refer readers to recent survey papers on the field for a more complete literature review (Shakya et al., 2023; Wang et al., 2022). However, we found that the update rule and regime proposed by (Watkins, 1989) were sufficient.

## 3. Methodology

**Notation.** Consider a dataset $D = \{(x_i, y_i)\}_{i=1}^n$, where each $x_i \in \mathbb{R}^d$ and $y_i \in \{0, 1\}$ pair is sampled i.i.d. from some unknown distribution $\mathcal{D}$. For the purposes of our work, these $x_i$'s may be continuous, ordinal, categorical, or binary. We refer to the $j^{th}$ *feature* of the $i^{th}$ *sample* as $x_{i,j}$. We use the notation $x_{\cdot,j}$ to refer to the $j^{th}$ feature. We consider binary classification problems in this work, though our results can be extended to multiclass classification with minor adjustments to the algorithms and theorems. We also work with *binarized* datasets, in which feature $j$ of $D$ is binarized into $B_j$ different *binary features*. For example, the feature $age$ may be binarized into binary features $age \leq 5$, $age \leq 10$, etc. We denote the $k^{th}$ binary feature corresponding to feature $j$ of the $i^{th}$ sample as $b_{i,j}^{(k)}$. We reserve capital letters $X$ and $Y$ for random variables, and we index random variables via subscripts, i.e., $X_i$.

Given a bit-vector $\theta \in \{0, 1\}^d$, we define the mask function $m(x_i, \theta) := (x_{i,j} \text{ if } \theta_j = 1; NA \text{ if } \theta_j = 0)_{j=1}^d$. For convenience, we often leave out dependence on $\theta$ and write $m(x_i)$ to denote a masked version of $x_i$. Let $J_{m(x_i)} := \{j | m(x_i)_j \in \mathbb{R}\}$. A *completion* of $m(x_i)$ is a vector $z \in \mathbb{R}^d$ s.t. $z_j = m(x_i)_j, \forall j \in J_{m(x_i)}$. When discussing cost sensitive optimization, we denote the cost associated with each input feature $x_{\cdot,j}$ as $c_j$.

### 3.1. Representing Trees to Resolve Predictive Equivalence

Given any decision tree $\mathcal{T}$, we represent $\mathcal{T}$ in a simplified disjunctive normal form (an OR of ANDs), which we denote by $\mathcal{T}_{\text{DNF}}$. See Figure 2 for an example of this representation.

This approach yields a number of useful properties. It remains globally interpretable, because we can present a simple logical formula for the whole tree. The new representation is still faithful to all the original predictions of the tree (Proposition 3.1). It can also make predictions whenever there is sufficient information to know the prediction on the original tree (Theorem 3.2), which we leverage later in our applications. It provides non-redundant explanations, meaning it does not suffer from the interpretability issues Izza et al. (2022) identify in decision trees (Proposition 3.3). It maps all predictively equivalent trees to the same form (The-

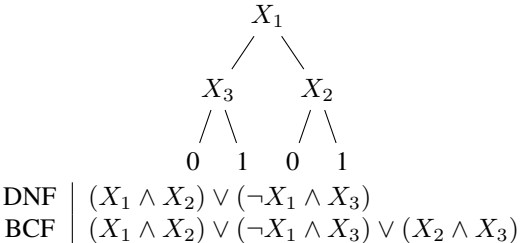

|     |     |
| --- | --- |
| DNF | $(X_1 \wedge X_2) \vee (\neg X_1 \wedge X_3)$ |
| BCF | $(X_1 \wedge X_2) \vee (\neg X_1 \wedge X_3) \vee (X_2 \wedge X_3)$ |

*Figure 2.* An example of a decision tree where the minimal DNF and Blake canonical forms differ. The minimal DNF of this tree describes the tree's behaviour with two cases. The Blake canonical form includes a third reason for predicting True, which always falls into the preceding two cases but relies on different variables.

orem 3.4). A proof for each of these statements is provided in Appendix A.

**Proposition 3.1** (Faithfulness). *Consider any tree $\mathcal{T}$ and let $x \in \mathbb{R}^d$ be a complete sample. Then $\mathcal{T}_{DNF}(x) = \mathcal{T}(x)$.*

**Theorem 3.2** (Completeness). *$\mathcal{T}_{DNF}(m(x)) \neq NA$ if and only if, for all completions $z$ of $m(x)$, $\mathcal{T}(z) = \mathcal{T}_{DNF}(m(x))$.*

**Proposition 3.3** (Succinctness). *Let the explanation for $\mathcal{T}_{DNF}(x)$ be any term in $SimplePosExpr$ that is satisfied by $x$ when $\mathcal{T}_{DNF}(x) = 1$ (or $SimpleNegExpr$ when $\mathcal{T}_{DNF}(x) = 0$). Then no variable in this explanation is redundant.*

**Theorem 3.4** (Resolution of Predictive Equivalence). *Decision trees $\mathcal{T}$ and $\mathcal{T}'$ are predictively equivalent if and only if $\mathcal{T}_{DNF} = \mathcal{T}'_{DNF}$ (with equality defined by Algorithm 3)*

Algorithm 1 describes how we transform trees into minimal DNF representation. This algorithm combines the positive-predicting leaves of the decision tree into an expression in disjunctive normal form. This expression is then simplified with a slightly modified version of the Quine-McCluskey algorithm (Quine, 1952) (see Algorithm 5) to find the minimal form of the boolean expression encoding positive predictions by the tree. We perform the same procedure on the negative-predicting leaves to obtain a minimal boolean expression for evaluating whether the tree predicts negative. Algorithm 2 explains how this method provides predictions, with 'substitute' meaning each variable with a known value is replaced by a constant (e.g., if $x_{i,1} = 1$, $(x_{\cdot,1} \wedge x_{\cdot,2}) \vee (\neg x_{\cdot,2})$ becomes $(1 \wedge x_{\cdot,2}) \vee (\neg x_{\cdot,2}) = True$). Equivalence is defined in Algorithm 3 in Appendix B.

While the basic simplified form has a number of useful properties, it does not directly afford all possible sufficient conditions for positive and negative predictions. Consider, for example, the tree in Figure 2: there are 3 sufficient conditions for a positive prediction, but our basic simplified form will only identify two of them. We leverage a second representation, called the Blake canonical form (Blake, 1937), to solve this problem: in Algorithm 4, we find all possible min-

---

**Algorithm 1** Compute DNF Representation from Tree

> **Input:** A decision tree $\mathcal{T}$.
> **Output:** $\mathcal{T}_{DNF}$, A minimal boolean formula in disjunctive normal form with equivalent logical form to $\mathcal{T}$.
> Let $L$ be the set of leaves of the decision tree, represented by a conjunction of the variables and decisions on the path to the leaf. Denote by $L^+$ the leaves that predict positive, and $L^-$ the leaves that predict negative.
> $PosExpr \leftarrow \vee_{l \in L^+} l$
> $NegExpr \leftarrow \vee_{l \in L^-} l$
> $SimplePosExpr \leftarrow QuineMcCluskey(PosExpr)$
> $SimpleNegExpr \leftarrow QuineMcCluskey(NegExpr)$
> Return $(SimplePosExpr, SimpleNegExpr)$

---

**Algorithm 2** Prediction with the DNF representation

> **Input:** $m(x)$, the sample to predict; $\mathcal{T}_{DNF}$.
> **Output:** Prediction from $\mathcal{T}_{DNF}(m(x))$ (0, 1 or NA)
> **for** term $t$ in $\mathcal{T}_{DNF}.SimplePosExpr$:
>    **Return** 1 if known feature values from $m(x)$ satisfy $t$
> **for** term $t$ in $\mathcal{T}_{DNF}.SimpleNegExpr$:
>    **Return** 0 if known feature values from $m(x)$ satisfy $t$
> expr $\leftarrow$ Substitute known feature values of $m(x)$ into $\mathcal{T}_{DNF}.SimplePosExpr$
> expr $\leftarrow QuineMcCluskey(\text{expr})$
> **Return** 1 if expr $== True$
> **Return** 0 if expr $== False$
> **Return** NA

imal sufficient conditions for a positive prediction, and all possible minimal sufficient conditions for a negative prediction. This also corresponds to identifying all partial concept classes (Alon et al., 2022) for which the tree predicts true (resp. False). This alternative form can optionally be used to simplify the prediction logic for our DNF – since it is now sufficient simply to evaluate each separate term in the DNF, without needing to do further logical simplification.

### 3.2. Datasets

We consider four datasets throughout this work and eight additional datasets in Appendix C. We refer to the primary four as COMPAS (Larson et al., 2016), Wine Quality (Cortez et al., 2009), Wisconsin (Street et al., 1993), and Coupon (Wang et al., 2017). COMPAS measures 7 features for 6,907 individuals, where labels are whether the individuals were arrested within 2 years of being released from prison. Wine Quality reports 11 features over 6,497 wines along with a numerical quality rating between 1 and 10. We binarize these ratings into high ($> 5$) and low ($\leq 5$) quality classes and predict this binary rating. Wisconsin is a breast cancer dataset and contains 30 features over 569 masses, where labels designate whether the tumor was malignant or benign. Coupon measures 25 features for 12,684 individuals, and

| Dataset | Total Trees | w/o Trivial | Ours |
|---------|-------------|-------------|------|
| COMPAS  | $12785 \pm 3e3$ | $3913 \pm 837$ | $2135 \pm 448$ |
| Coupon  | $666 \pm 54$ | $136 \pm 19$ | $55 \pm 8$ |
| Wine    | $6936 \pm 700$ | $2341 \pm 377$ | $1409 \pm 256$ |
| Wisc.   | $24052 \pm 9e3$ | $11990 \pm 5e3$ | $4657 \pm 2e3$ |

*Table 1.* Total number of trees, number of trees without trivial redundancies, and number of predictively nonequivalent trees (ours) in the Rashomon set. We abbreviate "Wine Quality" to "Wine" and "Wisconsin" to "Wisc."

labels denote whether or not the individual would accept a coupon. See Appendix D.1 for complete details on the preprocessing applied to each dataset.

## 4. Quantifying Predictive Equivalence

We can directly identify predictively equivalent decision trees using Algorithm 3. We now apply these tools to the Rashomon set of decision trees, found by the TreeFARMS algorithm, to measure the prevalence of predictive equivalence in practice (Xin et al., 2022). The Rashomon set is defined as the set of all models in a hypothesis space $\mathcal{F}$ within $\varepsilon$ training objective of the optimal model, where the objective is denoted $\mathrm{Obj}(f, D)$. Given an optimal model on the training data $f^* \in \arg\min_{f \in \mathcal{F}} \mathrm{Obj}(f, D)$, the Rashomon set is defined as:

$$\mathcal{R}(\mathcal{F}, D) := \{f \in \mathcal{F} | \mathrm{Obj}(f, D) \leq \mathrm{Obj}(f^*, D) + \varepsilon\}.$$

TreeFARMS uses a branch and bound algorithm with dynamic programming to find the Rashomon set of sparse decision trees, with $\mathcal{F}$ the hypothesis space of decision trees and $\mathrm{Obj}(f, D)$ defined as misclassification error plus a constant penalty for each leaf in the tree (Xin et al., 2022). The algorithm maintains a lower bound on the objective function of each possible subtree, and uses these lower bounds to prune large sections of the search space which provably cannot lead to near-optimal models.

We compute the Rashomon set of decision trees for the COMPAS, Coupon, Wine Quality, and Wisconsin datasets, and compare the total number of decision trees in each set to the number of unique DNF forms within each set. We use TreeFARMS (Xin et al., 2022) with maximum depth 3 and a standard per-leaf penalty of 0.01, identifying all trees within 0.02 of the optimal training objective. TreeFARMS can optionally remove trees that are trivially equivalent to other trees in the Rashomon set (i.e., the last split along some path leads to the same prediction in both leaves), so we also present the number of trees that have no such trivial splits. Going beyond trivial splits, we use our representation to identify the number of trees with unique decision logic. Table 1 presents this measure of Rashomon set size averaged over 5 folds of each dataset. We found that our represen-

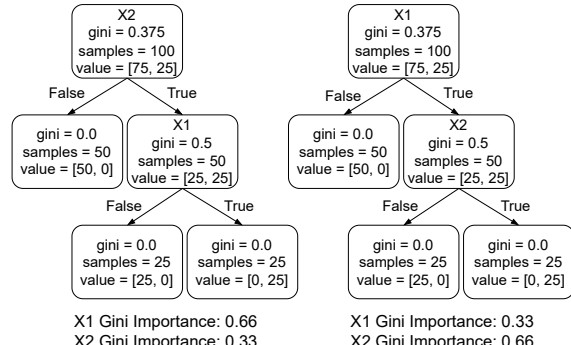

X1 Gini Importance: 0.66    X1 Gini Importance: 0.33
X2 Gini Importance: 0.33    X2 Gini Importance: 0.66

*Figure 3.* Two equivalent decision trees for the setting where $Y = X_1 X_2$ and $X_1, X_2 \overset{\text{i.i.d.}}{\sim} Bernoulli(0.5)$. Although they *always* produce identical predictions, achieve the same objective value, and are produced by the same algorithm, these two trees produce dramatically different variable importance values. `gini` refers to the Gini coefficient of each leaf, `samples` to the number of points falling into each leaf, and `value` denotes the number of negative (left) and positive (right) samples in the leaf.

tation revealed a substantial number of trees with identical decision logic. Appendix C.2 presents similar results across many Rashomon set parameter configurations.

## 5. Case Study 1: Variable Importance

### 5.1. Gini Importance

Predictive equivalence poses an immediate challenge for variable importance methods. To demonstrate this, we consider the toy setting where $Y = X_1 X_2$ and $X_1, X_2 \overset{\text{i.i.d.}}{\sim} Bernoulli(0.5)$. Figure 3 presents two distinct decision trees that perfectly match this data generating process. Even in this simple case, we observe that **equivalent trees can produce dramatically different variable importances** when computing an impurity-based variable importance such as Gini importance. The first tree claims $X_0$ is twice as important as $X_1$ and the second tree claims the opposite.

This effect becomes more pronounced with more variables. We next consider a similar data generating process with $Y = \prod_{i=1}^{10} X_i$ and $X_1, \ldots, X_{12} \overset{\text{i.i.d.}}{\sim} Bernoulli(0.5)$. There are 12 input variables but only variables 1 though 10 are used in the data generation process, meaning there are 2 unimportant variables. We greedily fit 3 predictively equivalent decision trees over the same data using different random seeds and measured the Gini importance of each variable to each tree. Figure 4 shows the distribution of importance for each variable over these trees. We observe that **the importance of each variable varies widely over predictively equivalent trees.** Moreover, the importance of some useful variables is nearly indistinguishable from

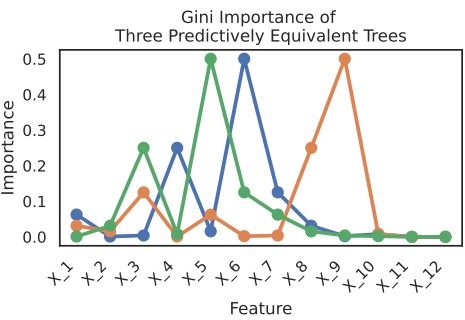

*Figure 4.* The Gini Importance for 12 variables over 3 predictively equivalent decision trees. Here, each color represents a different tree. Even though these trees are predictively equivalent, they produce radically different variable importance values.

the importance of the extraneous variables – e.g., $X_{10}$ has importance close to 0.

## 5.2. Rashomon Importance Distribution

We now examine the impact of predictive equivalence on a state-of-the-art variable importance method: the Rashomon Importance Distribution (RID, Donnelly et al., 2023). RID computes a stable cumulative density function (CDF) of variable importance over possible datasets using variable importance over the Rashomon set. In particular, the value of this CDF for feature $j$ at value $k$ (i.e., the probability that feature $j$ has importance less than or equal to $k$) is computed as the expected proportion of models in the Rashomon set for which feature $j$ has importance less than $k$.

RID is defined over a set of functions, meaning it expects each member of the Rashomon set to be a unique input-output mapping. In practice, however, RID operates over the Rashomon set of decision trees computed by TreeFarms (Xin et al., 2022). This set contains multiple predictively equivalent trees, which effectively places more weight on functions that can be expressed through many distinct trees and biases RID toward the variables that are important in these duplicated models. However, this bias can be removed by considering only one member of each set of predictively equivalent trees using our representation.

To demonstrate this effect, consider the following simple data generating process (DGP). With input variables $X_1, X_2 \sim Bernoulli(\sqrt{0.5})$ and $X_3 \sim Bernoulli(0.9X_1X_2+0.05)$, let $Y \sim Bernoulli(0.9X_3 + 0.05)$. We compute a "ground truth" variable importance value for this DGP by computing the permutation importance of each variable to the model $f(X_1, X_2, X_3) = X_3$.

Table 2 reports the 1-Wasserstein distance between the ground truth importance value and the distribution of impor-

| Method | Distance to Ground Truth | | |
|---|---|---|---|
| | $X_1$ | $X_2$ | $X_3$ |
| Original RID | 0.120 | 0.136 | 0.232 |
| PE Corrected RID | 0.092 | 0.105 | 0.182 |

*Table 2.* The 1-Wasserstein distance (Vaserstein, 1969; Kantorovich, 1960) between the ground truth importance value (represented as a distribution with all weight at the single true value) and the distribution from RID with and without correcting for predictively equivalent trees on the synthetic case described in Section 5.2. Controlling for predictive equivalence improves the estimated importance of each variable.

tance from RID with and without correcting for predictive equivalence. **When predictively equivalent trees are not accounted for, RID places more weight further from ground truth on all three variables**.

The confounding effect of predictively equivalent trees on RID can also be observed on real data. Figure 5 shows the distribution of importance from RID before and after controlling for predictively equivalent trees for three important variables on the COMPAS dataset. While there is no known ground truth importance value to compare against here, we see that a substantial distribution shift also occurs on real data. In fact, for each variable, the two-sample Kolmogorov-Smirnov test for the equality of distributions found a significant difference between distributions for each variable with a target p-value of 0.05, with test statistics of 0.043 for age, 0.048 for juvenile crimes, and 0.059 for priors count and $p < 0.001$ in each case. Appendix C.3 reports these values over additional datasets, and finds significant distribution shift in at least one variable for every dataset except one, for which a decision stump is sufficient.

## 6. Case Study 2: Missing Data

Decision trees are regularly used in the presence of missing data because they can be easily adjusted to handle missingness (Therneau et al., 1997). Our method allows identification of many cases where adjustments are not needed.

Consider a setting where data is missing from the test set, but there is no observed missingness in the training set. The standard approach is to impute missing features, but this threatens interpretability by complicating the pipeline from input data to prediction. Our representation can identify all cases where imputation is not needed across a wide range of missingness settings, avoiding this issue. Theorem 3.2 and its Corollary 6.1 establish that whenever $\mathcal{T}_{\text{DNF}}$ makes a non-NA prediction, the prediction matches the tree's prediction under perfect oracle imputation (meaning the oracle directly provides the missing value). As per Corollary 6.2, that means we can use DNFs to handle missingness in a way that is robust to a wide range of missingness mechanisms.

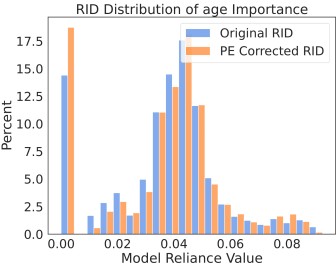 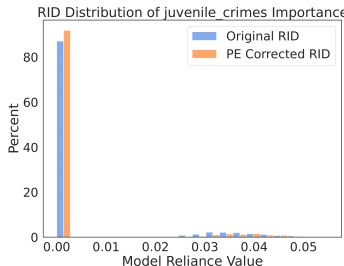 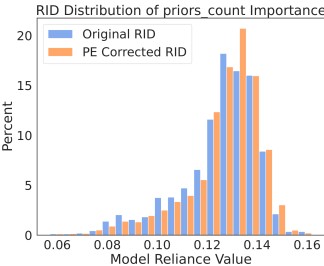

*Figure 5.* The distribution of variable importance from RID for three important variables on the COMPAS dataset, with and without correcting for predictive equivalence. When adjusting for predictive equivalence (shown in orange), more probability mass is given to zero importance for age and number of juvenile crimes, while high importance values receive more probability mass for number of priors. All other variables in this dataset had all probability mass at 0 importance in both cases.

The only setting where we may lose information is when missingness itself is informative about the label – but in a test-time missingness setting, where we have not seen any data with missingness during training, it is not possible to train a model to handle informative missingness anyway. Proofs of these corollaries are given in Appendix A.

**Corollary 6.1** (Irrelevance of Imputation). *Let* $x \in \mathbb{R}^d$. *Let* $g : \mathbb{R} \cup \{NA\}^d \to \mathbb{R}^d$ *be any imputation function. If* $\mathcal{T}_{DNF}(m(x)) \neq NA$, *then* $\mathcal{T}(g(m(x))) = \mathcal{T}(x)$, *which corresponds to oracle imputation.*

**Corollary 6.2** (Unbiasedness under test-time missingness). *Let* $x \in \mathbb{R}^d$. *When* $\mathcal{T}_{DNF}(m(x)) \neq NA$, *its predictions are an unbiased estimator for* $T(x)$ *with respect to the random missingness mechanism. This holds even if the mechanism is Missing Not At Random.*

We demonstrate empirically that decision trees rarely require additional missingness handling to predict on samples with missing data. In Figure 6, we introduce synthetic missingness (Missing Completely at Random) to a variety of real-world datasets by independently removing each feature of each sample with probability $p$. Using our DNF-based prediction method, we demonstrate that decision trees can regularly predict on a substantial number of points even when many features are missing. This means a decision tree's prediction is the same for most samples regardless of how a practitioner handles missing data, including any choice of imputation (Corollary 6.1).

In Figure 6, we show trees can predict substantially more often than standard ways of determining when a decision tree requires missingness handling would suggest. Since trees are ordinarily evaluated by following a path from root to leaf, the "path-based" baseline reports NA when a split is encountered that depends on a feature of unknown value. This same approach was recently used in a study of models' reliance on missingness-specific logic (Stempfle & Johansson, 2024). We also compare to a function-agnostic baseline that checks whether any feature of the model is missing. Ex-

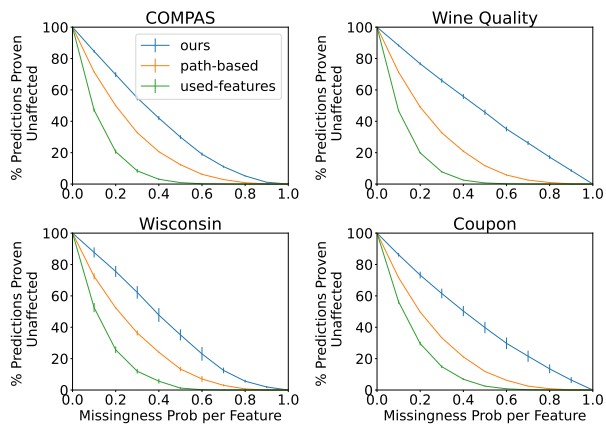

% Samples such that CART trees can Predict without Imputation

Improvement at 50% Missingness per Feature:

| | Compas | Wine | Wisconsin | Coupon |
|---|---|---|---|---|
| path | 2.4× | 3.9× | 2.6× | 3.4× |
| features | 32.8× | 64.6× | 28.6× | 16.1× |

*Figure 6.* Rate at which decision trees can make predictions as missing values are added. These results use a simple CART tree, as implemented by SKLearn (Pedregosa et al., 2011), with depth 3 and default parameters. The table shows the ratio of # samples identified under our method vs the two baselines.

periments on more datasets and with more tree algorithms are in Appendix C.4 and Appendix E, respectively.

We can extend this investigation of decision tree robustness beyond individual trees to the set of all near-optimal decision trees (the Rashomon set). We quantify how often a sample can be classified without using imputation by at least one decision tree in the Rashomon set (as found by TreeFARMS, Xin et al., 2022). We also show that when we use the best available model from the Rashomon set for each sample, we achieve comparable accuracy to the optimal model *if we had not had missing data*. Figure 7 shows that a majority of

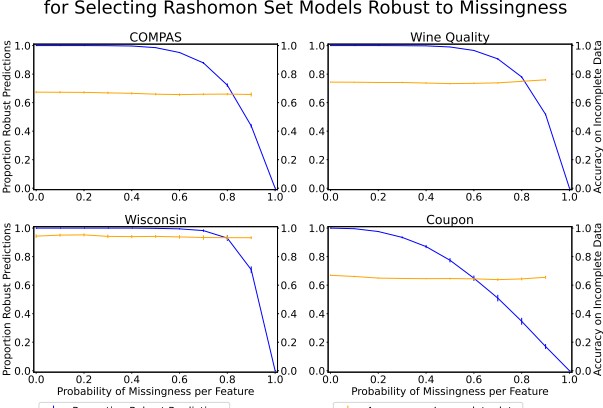

*Figure 7.* Rate at which at least one near-optimal tree can continue to make predictions as missing values are added, as well as accuracy on those predictions. These results use TreeFARMS (Xin et al., 2022) with maximum depth 3 and a standard per-leaf penalty of 0.01, with an epsilon of 0.02

samples can be predicted even with test-time missingness probability above 50% per feature. Note that the added ability to evaluate trees under the DNF form is built into the Rashomon set – if one tree is in the Rashomon set, all its predictively equivalent trees that are similar in sparsity and depth are also in the Rashomon set.

## 7. Case Study 3: Improving Cost Efficiency

When a user obtains the value for each feature in an "online setting" – i.e., iteratively decides which feature to discover – it may be tempting to simply traverse a decision tree and purchase features in the order they are encountered. If each feature has an associated cost, this naïve approach is needlessly expensive. We demonstrate that our decision tree simplification can reduce the cost of evaluating a tree *without changing the decision boundary at all*.

We introduce a Q-learning approach to learn the least expensive way to evaluate a decision tree. If any clause in the Blake canonical form of a decision tree is satisfied, we know sufficient information to form a prediction. Thus, the goal is to learn the minimum cost policy that satisfies at least one clause of this representation, yielding the following setting:

**State space**: Each state is defined by the status of all (binarized) features, where each feature may be 0, 1, or unknown. With $d_b := \sum_{j=1}^{d} B_j$ binary features, this yields $3^{d_b}$ states.

**Actions**: In each state, the Q-learner chooses to obtain one of the unknown features, transitioning to either the state where the measured feature is 0 or 1. For example, working

with the state $\{x_{\cdot,1} =?, x_{\cdot,2} =?\}$, purchasing $x_{\cdot,1}$ transitions to either $\{x_{\cdot,1} = 0, x_{\cdot,2} =?\}$ or $\{x_{\cdot,1} = 1, x_{\cdot,2} =?\}$. In each episode, a random row of the training dataset is selected, and the value in this row is used to determine the value of each queried feature. We restrict the actions available to the Q-learner to only include the features used in the current decision tree of interest.

**Reward Function.** When a feature $b_{i,j}^{(k)}$ (the $k^{th}$ bin on the $j^{th}$ feature of the $i^{th}$ example) is measured, a cost of $c_j$ (i.e., a reward of $-c_j$) is incurred if there is no $k'$ such that $b_{i,j}^{(k')}$ has been purchased; otherwise, no cost is incurred to reflect the fact that a practitioner would obtain the value of the input feature, not an individual bin. If enough features are known to satisfy any clause of the Blake canonical form of a tree, a reward of $\sum_{j=1}^{d} c_j$ is given, and the current episode of Q-learning is terminated.

Q-learning generally aims to learn a $num\_states \times num\_actions$ matrix, indicating the quality of every action in every state. In our setting, this yields a $3^{d_b} \times d_b$ matrix, which is infeasible to store – this matrix would have $4.23 \times 10^{28}$ entries on the largest of our datasets. We address this problem in two ways. First, we consider only "reasonable actions" – actions that measure a feature that is actually used in the tree, immediately ruling out any state related to measuring other features. Second, we avoid creating this large matrix by instead using a hash table that maps from a state to a $d_b-$dimensional vector indicating the expected reward of each action in that state. This hash table is initially empty; when a new state is visited during training, a new $d_b-$dimensional vector is added to the hash table. This procedure allows us to avoid storing information for states that are never realized – for example, if two binarized features signify $age < 5$ and $age < 8$, it is impossible for the former to be true while the latter is false.

We initialize our hash table using the reward obtained by directly traversing the decision tree of interest; Appendix G describes this procedure in detail. In our experiments, we run 10,000 episodes of exploration to train our Q-learner. After training, this yields a simple policy that recommends which feature to purchase in each state.

We evaluate the cost savings of this approach using the COMPAS, Wine Quality, Wisconsin, and Coupon datasets. For each dataset considered, we randomly generate an integer cost between 1 and 10 for each feature. We consider three purchasing policies: 1) following the BCF/Q-learning policy as outlined above, 2) purchasing features in the order suggested by traversing the tree, and 3) directly purchasing every feature in the tree. We then evaluate the average cost incurred by each policy across samples from the test dataset. We fit 50 decision trees on distinct bootstrap samples of each dataset, and perform this evaluation for each tree produced.

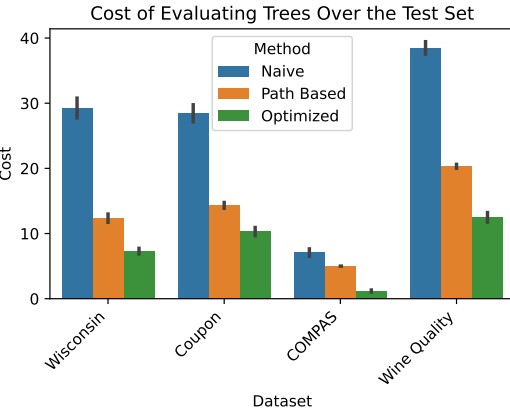

*Figure 8.* The cost of evaluating a tree by directly purchasing every feature in the tree (Naïve), purchasing features in the order suggested by traversing the tree (Path Based), and by following our BCF/Q-learning policy (Optimized). Error bars report standard deviation of cost over 50 trees, each learned from a different bootstrap of the original dataset.

Figure 8 shows the results of this evaluation over 50 trials. We find that **optimizing purchases based on our representation reduces the average cost of evaluating trees on every dataset**. Moreover, we see that purchasing features intelligently can dramatically reduce the cost of evaluating a tree relative to the naïve approach in which all features in the tree are purchased. It is important to note that this comes at no cost in terms of predictive accuracy, since the exact same decision boundary is applied in each case.

## 8. Conclusion

We proposed a simplified boolean logical representation of decision trees that decouples the logic encoded by a decision tree from its evaluation procedure. We showed that this approach can be used to account for predictive equivalence. In several case studies, we demonstrated the practical utility achieved by our representation. Future work could analyze the group structure of decision trees, where predictive equivalence is captured by equivalence classes defined by the trees' underlying logical models. It could also explore predictive equivalence's effects on tree ensembles.

## Acknowledgments

We acknowledge funding from the National Institutes of Health under 5R01-DA054994, the National Science Foundation under award NSF 2147061, and through the Department of Energy under grant DE-SC0023194. Additionally, this material is based upon work supported by the National Science Foundation Graduate Research Fellowship under Grant No. DGE 2139754. We thank Xenia Konti for her helpful conversations in developing Case Study 3.

## Impact Statement

This paper presents work whose goal is to advance the field of Machine Learning. There are many potential societal consequences of our work, none which we feel must be specifically highlighted here.

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

# A. Proofs

We start with a simple lemma that is useful in showing a number of the paper's propositions.

**Lemma A.1.** *Let $\mathcal{T}$ be a decision tree. Consider $\mathcal{T}_{DNF} := (SimplePosExpr, SimpleNegExpr)$ obtained from Algorithm 1. Let $x \in \mathbb{R}^d$ be a complete sample. $SimplePosExpr$ is satisfied by $x$ if and only if $\mathcal{T}(x) = 1$. Likewise $SimpleNegExpr$ is satisfied by $x$ if and only if $\mathcal{T}(x) = 0$.*

*Proof.* For a complete sample, we know $\mathcal{T}(x) \in \{0, 1\}$.

Denote by $L^+$ the leaves of $\mathcal{T}$ predicting 1, where each leaf is a logical expression encoding the path from root to leaf in $\mathcal{T}$. Denote $PosExpr := \vee_{\ell \in L^+} \ell$ to be the disjunction of all leaves predicting 1.

$\mathcal{T}(x) = 1$ if and only if one of its positive leaves $\ell \in L^+$ is satisfied by $x$. Since $PosExpr$ is a disjunction over the leaves of $\mathcal{T}$, $PosExpr$ is satisfied by $x$ if and only if $\mathcal{T}(x) = 1$. The Quine-McCluskey algorithm's output, $SimplePosExpr$, is logically equivalent to its input $PosExpr$. Thus, $SimplePosExpr$ is satisfied by $x$ if and only if $\mathcal{T}(x) = 1$.

Note that $NegExpr := \vee_{\ell \in L^-} \ell$ is the logical complement of $PosExpr$, because all samples fall into exactly one leaf and all leaves are exactly one of positive or negative. Also note that because Quine-McCluskey preserves logical equivalence, we know $SimplePosExpr$ is the logical complement of $SimpleNegExpr$. So $SimpleNegExpr$ is satisfied by a complete sample $x$ if and only if $SimplePosExpr$ is not satisfied by $x$. Thus, $SimpleNegExpr$ is satisfied by $x$ if and only if $\mathcal{T}(x) = 0$. $\square$

## A.1. Proof of Theorem 3.2

Consider a decision tree $\mathcal{T}$ and its corresponding minimal DNF representation $\mathcal{T}_{DNF}$. Let $x \in \mathbb{R}^d$ be a complete sample and consider an incomplete masked version $m(x)$. Then $\mathcal{T}_{DNF}(m(x)) = 1$ (resp. 0) if and only if all possible completions of $m(x)$ are classified as 1 (respectively 0) by $\mathcal{T}$.

*Proof.* We proceed by cases.

**Case 1: $\mathcal{T}_{\mathbf{DNF}}(m(x)) = NA$.** In this case, we will prove $\mathcal{T}$ does not make the same prediction on all completions of $m(x)$. We prove this by contradiction.

Suppose that, for all completions $z \in \mathbb{R}^d$ of $m(x)$, $\mathcal{T}(z) = \hat{y}$. WLOG assume $\hat{y} = 1$. Assume for contradiction that $\mathcal{T}_{DNF}(m(x)) = NA$. Since $\mathcal{T}_{DNF}(m(x)) = NA$, we must have that $SimplePosExpr$ cannot be satisfied by the variable assignments in $m(x)$. Therefore, there must be some completion $z$ of $m(x)$ so that $SimplePosExpr$ is falsified by $z$. By Lemma A.1, this implies $\mathcal{T}(z) = 0$, but this is a contradiction since $\hat{y} = 1$. Thus, if $\mathcal{T}_{DNF}(m(x)) = NA$, then $\mathcal{T}$ cannot make the same prediction on all completions of $m(x_i)$.

**Case 2: $\mathcal{T}_{\mathbf{DNF}}(m(x_i)) \neq NA$.** Suppose WLOG $\mathcal{T}_{DNF}(m(x)) = 1$. Then we know that the known variable assignments in $m(x)$ satisfy $SimplePoxExpr$. Therefore, for any completion $z$ of $m(x)$, $SimplePosExpr$ is satisfied by $z$. Thus, by Lemma A.1, we have $\mathcal{T}(x) = 1$ for all completions of $m(x)$. $\square$

## A.2. Proof of Proposition 3.1

**Proposition** (Faithfulness). Consider a decision tree $\mathcal{T}$ and a complete sample $x$. Then $\mathcal{T}(x) = \mathcal{T}_{DNF}(x)$.

*Proof.* Consider the special case of Theorem 3.2, where $m(x) = x$. We have $\mathcal{T}_{DNF}(m(x)) = \mathcal{T}_{DNF}(x) = \mathcal{T}(x)$. $\square$

## A.3. Proof of Proposition 3.3

**Proposition** (Succinctness). Let $x \in \mathbb{R}^d$. Assume $\mathcal{T}(x) = \hat{y} \in \{0, 1\}$, and let the explanation for $\mathcal{T}_{DNF}(x)$ be any term in $SimplePosExpr$ satisfied by $x$ if $\hat{y} = 1$, and any term in $SimpleNegExpr$ satisfied by $x$ if $\hat{y} = 0$. If any term in either expression is satisfied, no variable in the explanation is redundant.

*Proof.* WLOG assume $\hat{y} = 1$. Let the explanation given by $\mathcal{T}_{DNF}(x)$ be any term in $SimplePosExpr$ satisfied by $x$. This explanation is sufficient to guarantee that the predict algorithm (Algorithm 2) will return 1, since that algorithm returns 1 if any term in $SimplePosExpr$ is satisfied by $x$. The explanation is non-redundant because each term in the output of the

QuineMcCluskey algorithm is a prime implicant of the input formula(Quine, 1952), and thus no subset of the literals in any term will guarantee satisfaction of the term. □

### A.4. Proof of Theorem 3.4

**Theorem** (Resolution of Predictive Equivalence). $\mathcal{T}_{\text{DNF}} = \mathcal{T}'_{\text{DNF}}$ if and only if $\mathcal{T}$ and $\mathcal{T}'$ are predictively equivalent.

*Proof.* We consider three cases: 1) the case where $\mathcal{T}$ and $\mathcal{T}'$ are not predictively equivalent; 2) the case where $\mathcal{T}$ and $\mathcal{T}'$ are predictively equivalent and use exactly the same set of input features; and 3) the case where $\mathcal{T}$ and $\mathcal{T}'$ are predictively equivalent and do not use exactly the same set of input features.

**Case 1.** First note that if $\mathcal{T}$ and $\mathcal{T}'$ are not predictively equivalent, the two trees cannot be logically equivalent by the definition of predictive equivalence. As such, they cannot have the same minimal DNF form, and cannot be equivalent as defined in Algorithm 3, since $set(\mathcal{T}.simplePosExpr) \neq set(\mathcal{T}'.simplePosExpr)$.

**Case 2.** Now consider the case where $\mathcal{T}$ and $\mathcal{T}'$ are predictively equivalent, and the set of features used in $\mathcal{T}$ matches the set of features used in $\mathcal{T}'$. Then truth table $T$ as used when simplifying $\mathcal{T}$ is identical to the truth table used when simplifying $\mathcal{T}'$. Since Algorithm 5's output is fully determined given the truth table, we know the output is the same for both trees: that is, $set(\mathcal{T}.simplePosExpr) = set(\mathcal{T}'.simplePosExpr)$ and therefore $\mathcal{T}_{\text{DNF}} = \mathcal{T}'_{\text{DNF}}$.

**Case 3.** Finally, we turn to the case when two trees are predictively equivalent and use a distinct set of features. In this case, the features that aren't shared across both trees are completely irrelevant to the trees' predictions. So we know no prime implicant can contain any of those feature values. Therefore, the set of prime implicants is the same when Quine McCluskey runs on each tree.

We then remove the columns corresponding to features that do not occur in any prime implicant, and only preserve rows that have 0 values for all of those features. At this point, both truth tables will be the same: they will both have the same columns since the set of prime implicants is the same. They will also have the same rows in the same order: when we only consider the set of truth table rows which have 0 values for all the irrelevant features that are not in any prime implicant, these rows will have the same relative ordering across the two tables , and they will both cover exactly the set of all possible assignments for features that are in at least one prime implicant.

Since the truth tables are the same and the set of prime implicants is the same, Quine-McCluskey gives the same output for these two trees in this case, and we will have $\mathcal{T}_{\text{DNF}} = \mathcal{T}'_{\text{DNF}}$.

Having covered all possible cases, we know that $\mathcal{T}_{\text{DNF}} = \mathcal{T}'_{\text{DNF}}$ if and only if $\mathcal{T}$ and $\mathcal{T}'$ are predictively equivalent. □

### A.5. Proof of Corollary 6.1

**Corollary** (Irrelevance of Imputation). Let $x \in \mathbb{R}^d$ be a complete sample with an incomplete masked version $m(x)$. Consider a decision tree $T$ and its DNF representation $\mathcal{T}_{\text{DNF}}$. Let $z$ be a completion of $m(x)$, obtained via any possible imputation method. Then, if $\mathcal{T}_{\text{DNF}} \neq NA$, $\mathcal{T}(z) = \mathcal{T}(x)$, where $\mathcal{T}(x)$ corresponds to the prediction of $\mathcal{T}$ with Oracle imputation on $m(x)$.

*Proof.* By Lemma A.1, we know that $\mathcal{T}_{\text{DNF}}(m(x)) = \hat{y} \in \{0, 1\}$ if and only if $T(z) = \hat{y}$, for all completions $z$ of $m(x)$. The set of all completions of $m(x)$ includes the original values of $x$, in addition to any possible imputation of $m(x)$. Therefore, for all imputations $z$ of $m(x)$, $\mathcal{T}(z) = \mathcal{T}(x)$. □

### A.6. Proof of Corollary 6.2

**Corollary** (Unbiasedness under test-time missingness). Let $x \in \mathbb{R}^d$ be a complete sample with an incomplete masked version $m(x, \theta)$ for some random missingness mask parameter $\theta \in \{0, 1\}^d$. When $\mathcal{T}_{\text{DNF}}(m(x, \theta)) \neq NA$, $\mathcal{T}_{\text{DNF}}(m(x, \theta))$ is an unbiased estimator over random draws of missingness for $T(x)$. This holds even if data is Missing Not At Random.

*Proof.* Let $\theta \in \{0, 1\}^d$ be a random variable drawn from some unknown distribution $\Theta$. Given a sample $x$, Theorem 6.1 tells us that when $\mathcal{T}_{\text{DNF}}$ can predict a non-NA value, that prediction matches the prediction of $\mathcal{T}$ without any missingness.

With this, we have that $\mathcal{T}_{\text{DNF}}(m(x,\theta))$ is an unbiased estimator of $\mathcal{T}(x)$ when it can make predictions:

$$\mathbb{E}_{\theta \sim \Theta : \mathcal{T}_{\text{DNF}}(m(x,\theta)) \neq NA} \left[ \mathcal{T}_{\text{DNF}}(m(x,\theta)) \right] = \mathbb{E}_{\theta \sim \Theta : \mathcal{T}_{\text{DNF}}(m(x,\theta)) \neq NA} \left[ \mathcal{T}(x) \right] \qquad \text{By Theorem 6.1}$$

$$= T(x) \qquad \text{Expectation of a constant}$$

That is, whenever $\mathcal{T}_{\text{DNF}}(m(x,\theta)) \neq NA$, $\mathcal{T}_{\text{DNF}}(m(x,\theta))$ is an unbiased estimator of $\mathcal{T}(x)$ as required. $\qquad \square$

## B. Algorithm Details

---

**Algorithm 3** Equality Checking

---

**Input:** $\mathcal{T}_{\text{DNF}}^{(1)}$, $\mathcal{T}_{\text{DNF}}^{(2)}$, the two trees to compare.
**Output:** True if and only if the two inputs are predictively equivalent.
$T_1 \leftarrow$ terms in $\mathcal{T}_{\text{DNF}}^{(1)}.SimplePosExpr$
$T_2 \leftarrow$ terms in $\mathcal{T}_{\text{DNF}}^{(2)}.SimplePosExpr$
Return $set(T_1) == set(T_2)$

---

**Algorithm 4** BCF

---

**Input:** $T$: a list of all terms in a minimal DNF Tree leading to a specific prediction (either positive or negative). (Corresponds to Blake Canonical Form)
**Output:** A list of all possible sufficient conditions for that specific prediction
Let $P$ be a list of all pairs (q, p) of distinct terms in $T$
**repeat**
  $(q,p) \leftarrow P.\text{pop}()$
  **if** there exists exactly one literal $z$ s.t $z \in q$ and $\neg z \in p$ **then**
    $q' \leftarrow q \setminus \{z\}$
    $p' \leftarrow p \setminus \{\neg z\}$
    **if** $q' \wedge p'$ is a contradiction or $q' \wedge p' \in P$ **then**
      Continue
    **else**
      **for** $t \in T$ **do**
        $P.\text{append}((q' \wedge p', t))$
      **end for**
      $T.\text{append}(q' \wedge p')$
    **end if**
  **end if**
**until** $P = \emptyset$

---

---

**Algorithm 5** Adjusted Quine-McCluskey (with particular processing of the data structures involved to allow for proof of Theorem 3.4)

---

**Input:** $D$: a DNF equation where each variable has name feature_$j$ for some integer $j$

**Output:** A logically equivalent DNF equation corresponding to a minimal set of prime implicants.

Create truth table T for expression $D$, where the columns are in order of variable name $j$, and rows are in order of boolean value (i.e., the first row is all variables False, then the rightmost variable True and all other variables False, and the last is all variables True)

$P \leftarrow$ all prime implicants of $T$

$C \leftarrow$ All columns corresponding to variables in $D$ that are not in any prime implicant.

Remove from $T$ all rows with a nonzero value in any column in $C$

Remove from $T$ all columns in $C$

Sort the prime implicants $p \in P$ based on the earliest-appearing row of $T$ that satisfies the implicant.

Find and return a minimal cover deterministically given $T$ and $P$. (Where cover corresponds to a set of prime implicants such that all rows satisfy at least one prime implicant)

---

Note that removing the columns in $C$ and the rows with nonzero values in these columns still preserves the validity of the output. Those columns are irrelevant to whether or not any prime implicant covers any particular row, since they never occur in any prime implicant. Further, covering the rows with 0 values in those columns also corresponds to covering rows with nonzero values in those columns, so we need not consider those additional rows when computing a cover.

---

## C. Experiments With Additional Datasets

In this section, we repeat each of the primary case studies from the main body of this work over eight additional datasets.

### C.1. Overview of Additional Datasets

We consider eight additional datasets, which we refer to as Netherlands (Tollenaar & van der Heijden, 2013), Broward (Wang et al., 2023), FICO (FICO et al., 2018), Spiral (McTavish et al., 2022), Tic-Tac-Toe (Aha, 1991), and Iris Setosa/Versicolor/Virginica (Fisher, 1936). Netherlands measures 9 features for 20,000 individuals, where labels are whether the individuals from the Netherlands committed another crime after being released from prison. Similarly, Broward reports 38 features from 1,955 individuals from Broward county, Florida, where labels are whether the individuals from the committed another crime after being released from prison. FICO contains 23 features from 10,459 individuals, and labels whether each individual will repay a line of credit within 2 years. In our experiments, we remove all rows from FICO that contain missing data, resulting in 2,502 samples. Spiral is a synthetic dataset with 2 features over 100 samples, where each sample is randomly drawn from one of two interweaving spirals and the features are the x and y coordinates of the sample. In spiral, labels indicate which spiral each sample was drawn from. Tic-Tac-Toe measures 9 features over all 958 possible games of tic-tac-toe, and labels each game as 1 if the first player won and 0 otherwise. The three Iris datasets – Iris Setosa, Iris Versicolor, and Iris Virginica – are different tasks generated from the same multiclass classification dataset. The original Iris dataset reports 4 features over 150 iris flowers, and labels each sample as Setosa, Versicolor, or Virginica. We transform this dataset into three one versus all binary classification datasets. For several appendix experiments, we also consider a ninth dataset, Higgs (Baldi et al., 2014), subsampled to a 1-million sample version. This dataset is too large for feasibly finding a full Rashomon set, so we only use this dataset for appendix replications of the single tree missing data results and the cost-sensitive results.

### C.2. Additional Results Quantifying Predictive Equivalence

Table 3 provides the size of the Rashomon set before and after correcting for predictive equivalence on our additional datasets. We find that, in all cases except Iris-Setosa, controlling for predictive equivalence cuts the size of the Rashomon set by at least half.

| Dataset | Total Trees | w/o Trivial | Ours |
|---|---|---|---|
| FICO | $8086 \pm 2202$ | $1969 \pm 643$ | $1154 \pm 375$ |
| Netherlands | $926 \pm 399$ | $265 \pm 111$ | $114 \pm 48$ |
| Spiral | $58 \pm 14$ | $38 \pm 13$ | $20 \pm 6$ |
| Tic-Tac-Toe | $354 \pm 158$ | $196 \pm 77$ | $72 \pm 29$ |
| Iris-Virginica | $196 \pm 124$ | $72 \pm 41$ | $44 \pm 25$ |
| Iris-Versicolor | $168 \pm 72$ | $73 \pm 26$ | $28 \pm 10$ |
| Iris-Setosa | $2 \pm 0$ | $2 \pm 0$ | $2 \pm 0$ |
| Broward | $4242 \pm 1165$ | $1741 \pm 487$ | $899 \pm 284$ |

*Table 3.* Total number of trees, number of trees without trivial redundancies, and number of predictively nonequivalent trees (ours) in the Rashomon set.

## C.3. Additional Results for Case Study 1: Variable Importance

In this section, we evaluate the shift in RID when controlling for predictive equivalence across all variables from every dataset considered in this work. For each dataset and variable, we computed RID using the parameters described in Section D with predictively equivalent trees included, and with all but one tree from each predictively equivalent set removed. We perform a Kolmogorov-Smirnov test to determine whether each pair of resulting distributions are significantly different. Tables 4 and 5 report the maximum distance between each of the two empirical distributions, as well as the p-value for the relevant Kolmogorov-Smirnov test. We find that, on every dataset except Iris Setosa, at least one variable exhibits significant distribution shift at $p < 0.05$. Figures 9 and 10 present the distribution from RID for each variable with a significant distribution shift. Note that we do not include variables that received zero importance from both methods in these tables, filtering out 148 of 217 total variables.

| Dataset | Variable | Sup. Distance | p-Value |
|---|---|---|---|
| Broward | p_fta_two_year | 0.011583 | 0.516924 |
| | three_year | 0.037849 | 0.000001 |
| | one_year | 0.009685 | 0.738680 |
| | six_month | 0.009828 | 0.722117 |
| | p_dui | 0.010082 | 0.692329 |
| | p_arrest | 0.020327 | 0.033135 |
| | p_misdemeanor | 0.049235 | 0.000000 |
| | age_at_current_charge | 0.006845 | 0.973439 |
| | age_at_first_charge | 0.028369 | 0.000685 |
| | p_charges | 0.025376 | 0.003370 |
| | p_pending_charge | 0.008679 | 0.847098 |
| COMPAS | age | 0.043373 | 0.000016 |
| | priors_count | 0.067886 | 0.000000 |
| | juvenile_crimes | 0.048217 | 0.000001 |
| FICO | NetFractionRevolvingBurden | 0.015801 | 0.999146 |
| | ExternalRiskEstimate | 0.079168 | 0.002619 |
| | NumSatisfactoryTrades | 0.025707 | 0.867512 |
| | MSinceMostRecentInqexcl7days | 0.105930 | 0.000014 |
| Iris Setosa | petal length | 0.085000 | 0.466286 |
| | petal width | 0.030000 | 0.999992 |
| Iris Versicolor | petal length | 0.145259 | 0.000417 |
| | petal width | 0.172575 | 0.000013 |
| Iris Virginica | petal length | 0.079877 | 0.004517 |
| | petal width | 0.080317 | 0.004225 |
| Netherlands | >20 previous case | 0.220336 | 0.000000 |
| | age at first penal case | 0.234413 | 0.000000 |
| | log # of previous penal cases | 0.154640 | 0.000000 |
| | 11-20 previous case | 0.156908 | 0.000000 |
| Spiral | feat1 | 0.056763 | 0.000218 |
| | feat2 | 0.046307 | 0.004580 |

*Table 4.* The amount of distribution shift in RID when controlling for predictive equivalence across all datasets considered in this work. We do not report results for variables that receive zero importance under either method. Continued in Table 5.

| dataset_nice | var | ks_test_stat | ks_test_p |
|---|---|---|---|
| Tic-Tac-Toe | Feat5_x | 0.015350 | 0.596928 |
| | Feat6_x | 0.032067 | 0.011964 |
| | Feat8_x | 0.043873 | 0.000140 |
| | Feat6_o | 0.021761 | 0.188094 |
| | Feat7_x | 0.010089 | 0.959806 |
| | Feat8_o | 0.019266 | 0.311956 |
| | Feat5_o | 0.007327 | 0.999217 |
| | Feat7_o | 0.007696 | 0.998273 |
| | Feat4_x | 0.125572 | 0.000000 |
| | Feat3_o | 0.005579 | 0.999998 |
| | Feat3_x | 0.010545 | 0.942370 |
| | Feat2_x | 0.062774 | 0.000000 |
| | Feat2_o | 0.021888 | 0.182977 |
| | Feat1_x | 0.008896 | 0.988349 |
| | Feat1_o | 0.005991 | 0.999989 |
| | Feat0_x | 0.062457 | 0.000000 |
| | Feat0_o | 0.017226 | 0.448192 |
| | Feat4_o | 0.081569 | 0.000000 |
| Wine Quality | alcohol | 0.028608 | 0.002097 |
| | volatile_acidity | 0.020111 | 0.067145 |
| | citric_acid | 0.002099 | 1.000000 |
| | free_sulfur_dioxide | 0.001559 | 1.000000 |
| Wisconsin | area2 | 0.004952 | 0.902248 |
| | area3 | 0.011916 | 0.047488 |
| | concave_points1 | 0.007035 | 0.531103 |
| | concave_points3 | 0.029109 | 0.000000 |
| | concavity3 | 0.007328 | 0.478215 |
| | perimeter3 | 0.010055 | 0.139228 |
| | radius3 | 0.006122 | 0.705633 |
| | texture1 | 0.003626 | 0.994981 |
| | texture3 | 0.022322 | 0.000004 |

*Table 5.* The amount of distribution shift in RID when controlling for predictive equivalence across all datasets considered in this work. We do not report results for variables that receive zero importance under either method. Continued from Table 4.

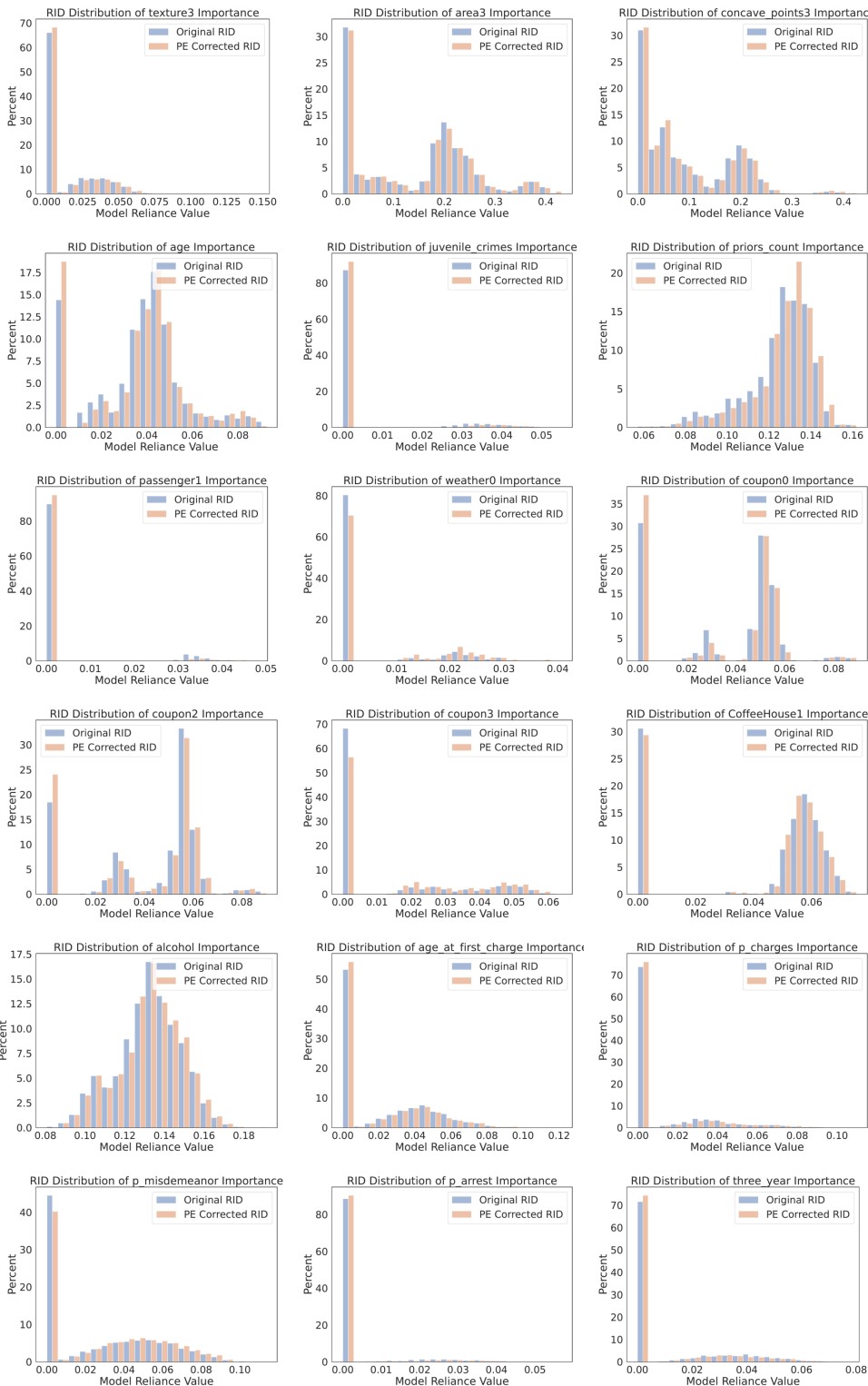

*Figure 9.* Importance distribution before (in blue) and after (in orange) controlling for predictive equivalence across all variables where RID showed significant distribution shift. Continued in Figure 10.

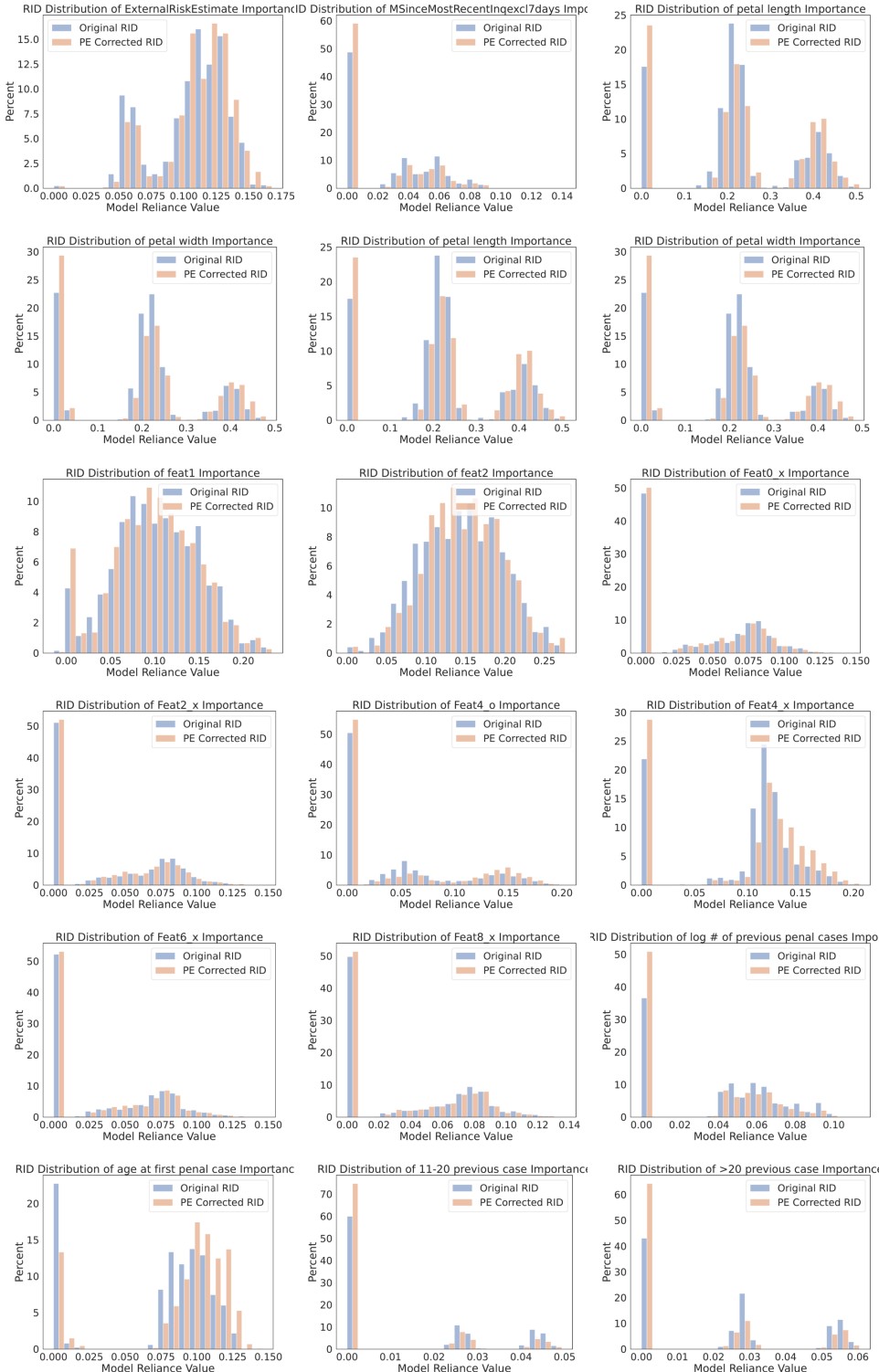

*Figure 10.* Importance distribution before (in blue) and after (in orange) controlling for predictive equivalence across all variables where RID showed significant distribution shift. Continued from Figure 9.

*Figure 11.* Rate at which decision trees can make predictions as missing values are added. These results use a simple CART tree, as implemented by SKLearn (Pedregosa et al., 2011), with depth 3 and default parameters.

### C.4. Additional Results for Case Study 2: Missing Data

Figures 11 and 12 include results on additional datasets. Results are similar to the main paper figures 6 and 7, with the exception of iris-setosa (where the tree found for each fold is usually a trivial depth 1 decision stump and exhibits no predictive equivalence).

## C.5. Additional Results for Case Study 3: Improving Cost Efficiency

We repeat the experimental setup from Case Study 3 for each of the additional datasets over 50 trials; the results of this evaluation are presented in Figure 13. We find that combining Q-learning with our tree representation **allows us to improve the cost of forming a prediction on every dataset but one**. On the Iris Setosa dataset, each method yields the same cost because the tree is simply a decision stump, meaning there is only one variable to purchase.

# D. Experimental Details

## D.1. Dataset Preprocessing

For Section 5.2 and Section C.3, we used the internal binarization procedure of RID (Donnelly et al., 2023), which binarizes according to Threshold Guessing from (McTavish et al., 2022), using the thresholds selected based on 40 boosted decision stumps.

For the Missing Data Section, we split datasets into 5 folds, and binarized according to Threshold Guessing from (McTavish et al., 2022), using the thresholds selected based on 40 boosted decision stumps. Standard errors reported are the standard deviation across folds divided by the square root of the number of folds.

Most datasets contained no pre-existing missing data; when they did, however (for the COMPAS dataset and Coupon) we removed rows with missing data in preprocessing.

For our experiments on cost sensitive optimization, we performed quantile-based binning on each feature using two quantiles per feature (the 0.33 quantile and the 0.66 quantile).

## D.2. Hyperparameter Settings

In all of our experiments with RID, we computed RID over 100 bootstrap iterations with an additive Rashomon bound of 0.02, a sparsity penalty of 0.02, and a maximum tree depth of 3 (i.e., the $depth\_bound$ parameter was set to 4). Note that, when using RID, we applied a larger sparsity penalty than we did in our other experiments (0.02 instead of 0.01) because we ran into computational constraints when running RID over very large Rashomon sets. We did not include "trivial extension" in these Rashomon sets because that is the default behavior of RID.

In our missing data experiments, we fit decision trees using SKLearn's Decision Tree implementation (Pedregosa et al., 2011), with a maximum depth of 3 and all other parameters left as their default values. Our Rashomon sets used depth 3 (i.e., the $depth\_bound$ parameter was set to 4), additive Rashomon bound 0.02, and sparsity penalty 0.01. (we did not include trivial extensions because those are irrelevant to what we were investigating; a trivial extension never allows handling of additional missing data). Missingness is injected into complete datasets synthetically. For each sample, we introduced missingness independently into each binarized feature with probability $p \in [0.1, 0.2, \ldots, 0.9]$. We investigate alternative depths of sklearn and pre-binarization missingness in Appendix Section E.

In our cost sensitive optimization experiments, we fit decision trees using SKLearn's Decision Tree implementation (Pedregosa et al., 2011) with a maximum depth of 3, and all other parameters left as their default values. In our Q-learner, we used a discount factor of 0.9, a learning rate of 0.1, and an exploration rate of 0.5, with each term defined as in (Watkins, 1989).

# E. Additional Missing Data Results

## E.1. Pre-binarization Missingness

The main paper presents results for a simple way of introducing MCAR missingness: each binary feature has an independent chance of being missing with probability $p \in [0, 1]$. However, we might also consider MCAR missingness injected in the original feature space, pre-binarization. Here each feature is still missing with probability $p$, but binary features from the same original feature are not independently missing - they must either all be missing or have none of them missing.

We reproduce the main paper's results with this alternative form of missingness in Figures 14 and 15. The number of cases where trees can still make predictions is diminished to some extent, but there remains a substantial proportion of trees completely robust to missingness, as identified with our method as opposed to the baselines, and as identified within an entire rashomon set.

## E.2. Robustness to Missingness on Near-Optimal Trees

We reproduce Figure 6 with optimal trees - GOSDT (Lin et al., 2020) and dl85 (Aglin et al., 2020) in Figure 16. We find that, for each of these trees, we can predict on substantially more samples than the baseline methods.

## E.3. Results at different depths

Figure 17 shows results from Figure 6 across different potential depths of decision tree for sklearn. The left column corresponds to results from the main paper, with depth 3. Deeper decision trees lead to similar conclusions. Two instances - COMPAS and Wine Quality - did not terminate within 12 hours, so for these two datasets we used a slightly relaxed version of the simplification procedure for our method (not running the Quine-McCluskey simplification step on instances with more than 8 variables), giving a lower bound on the number of unaffected predictions (the baselines remained unchanged).

# F. Additional Cost Sensitive Optimization Results

### F.1. Cost Optimization With an Additional Baseline

In the main paper, we presented three methods of evaluating a decision tree in a cost sensitive setting: a naive approach, in which every variable used by the decision tree is purchased; a path-based approach, in which variables are purchased as they are encountered while traversing a path in a decision tree; and our BCF based Q-learning approach, in which we aim to satisfy a clause in the BCF of the tree as cheaply as possible. Here, we aim to disentangle how much of the observed gain in cost efficacy was due to the Q-learning approach, and how much could be directly obtained through the BCF.

To do so, we introduce another method to our cost-sensitive evaluation in which we iteratively purchase the cheapest unknown feature used by the tree, and check whether any clause in the BCF has been satisfied. This approach leverages the early stopping enabled by BCF, but follows a simple heuristic instead of a learned policy. We refer to this approach as "Greedy".

Figure 18 presents the results of this evaluation across all datasets considered. We observe that, as expected, our Q-learning based approach achieves equal or better performance than this baseline on every dataset. We also observe that this greedy approach sometimes obtains substantially better performance than path-based traversal, but is very inconsistent – in some cases, it is almost as inefficient as the naïve baseline. As such, we conclude that the Q-learning component of our approach is necessary to reliably reduce cost.

### F.2. Hyperparameter Analysis for Q-Learning

In the main paper, we presented the cost of traversing trees using our BCF based Q-learning approach with one particular set of hyperparameters. Here, we aim to evaluate the sensitivity of these results to our hyperparameter selection.

Across the four primary datasets evaluated in the main paper, we consider three different settings for three important hyperparameters: alpha, gamma, and the exploration rate. Figure 19 presents the results of this evaluation. We observe that, across reasonable hyperparameter settings, our BCF Q-learning framework produces almost identical results. This suggests that the method is in fact converging, and our analysis is not substantially impacted by the hyperparameters we chose.

### F.3. Cost Optimization on Cost-Optimal Trees

When evaluating our cost optimization approach, we have primarily considered decision trees that were optimized for predictive accuracy alone. Here, we instead start with cost-optimal decision trees as produced by STreeD (van der Linden et al., 2023). We consider the same evaluation methods and datasets as in prior experiments.

Figure 20 shows the results of this evaluation across different levels of priority given to cost minimization. We find that our Q-learning approach never produces a substantial increase in evaluation cost, although in some settings we observe a very slight increase.

Surprisingly, we also observe some cases (e.g., Wine Quality, Wisconsin, and Iris Virginica for weight 0.0001) where we can evaluate a "cost-optimal" decision tree in a cheaper way than directly traversing the tree. This reveals an interesting nuance. The trees produced by (van der Linden et al., 2023) are guaranteed to be optimal with respect to a given cost-accuracy tradeoff, where cost corresponds to the cost of variables when predicting with a tree from the top down, and trees are subject to depth and sparsity constraints. These constraints are used to ensure the classifier can be represented as an interpretable, small tree, and help with computation time. However, a given tree can have predictively equivalent forms that are deeper and/or less sparse, but yield lower average cost. In effect, our Q-learning framework allows us to use one predictively equivalent form to communicate the tree simply, and a less-constrained predictively equivalent *evaluation procedure* for the tree that is completely faithful to the other form while being more cost-effective.

## G. Detailed Description of Q-learner Initialization

In this section, we describe how we use direct traversal of the target decision tree to initialize our Q-learner. Algorithm 6 provides pseudo-code describing the procedure; at a high level, we recursively traverse the decision tree until a leaf is reached. We pay the cost of each feature purchased along the way, and compute the reward at a given node as the weighted average of the reward of that node's two child nodes, where the weight is determined by the empirical probability of taking each path across the training dataset.

---

**Algorithm 6** Initialization of Q-Learner Using Path Traversal

---

**Input:** $\mathcal{T}$, the tree to optimize for cost; $D$, the training split of the dataset; $Q$, the (empty) hash table for our Q-learner; $r$, the reward given when a prediction is possible; $d$, the number of possible actions.

**function** initialize($\mathcal{T}_{cur}, D_{cur}, state$)

    **if** $\mathcal{T}_{cur}$ is a leaf node **then**

        **return** $r$

    **else**

        $j \leftarrow \mathcal{T}_{cur}.next\_split$

        $D_\ell \leftarrow$ subset of $D_{cur}$ where $x_{\cdot,j} = 0$
        $p_\ell \leftarrow$ proportion of $D_{cur}$ where $x_{\cdot,j} = 0$
        $state_\ell \leftarrow$ copy of $state$ with feature $j$ set to 0
        $r_\ell \leftarrow$ initialize($\mathcal{T}_{cur}.left\_subtree, D_\ell, state_\ell$)

        $D_r \leftarrow$ subset of $D_{cur}$ where $x_{\cdot,j} = 1$
        $p_r \leftarrow$ proportion of $D_{cur}$ where $x_{\cdot,j} = 1$
        $state_r \leftarrow$ copy of $state$ with feature $j$ set to 1
        $r_r \leftarrow$ initialize($\mathcal{T}_{cur}.right\_subtree, D_r, state_r$)

        $\bar{r} \leftarrow p_\ell r_\ell + p_r r_r - cost(j)$
        $Q[state] \leftarrow \mathbf{0} \in \mathbb{R}^d$
        $Q[state][j] \leftarrow \bar{r}$

        **return** $\bar{r}$

    **end if**

**end function**

$state \leftarrow$ initial state with all features unknown
initialize($\mathcal{T}, D, state$)

---

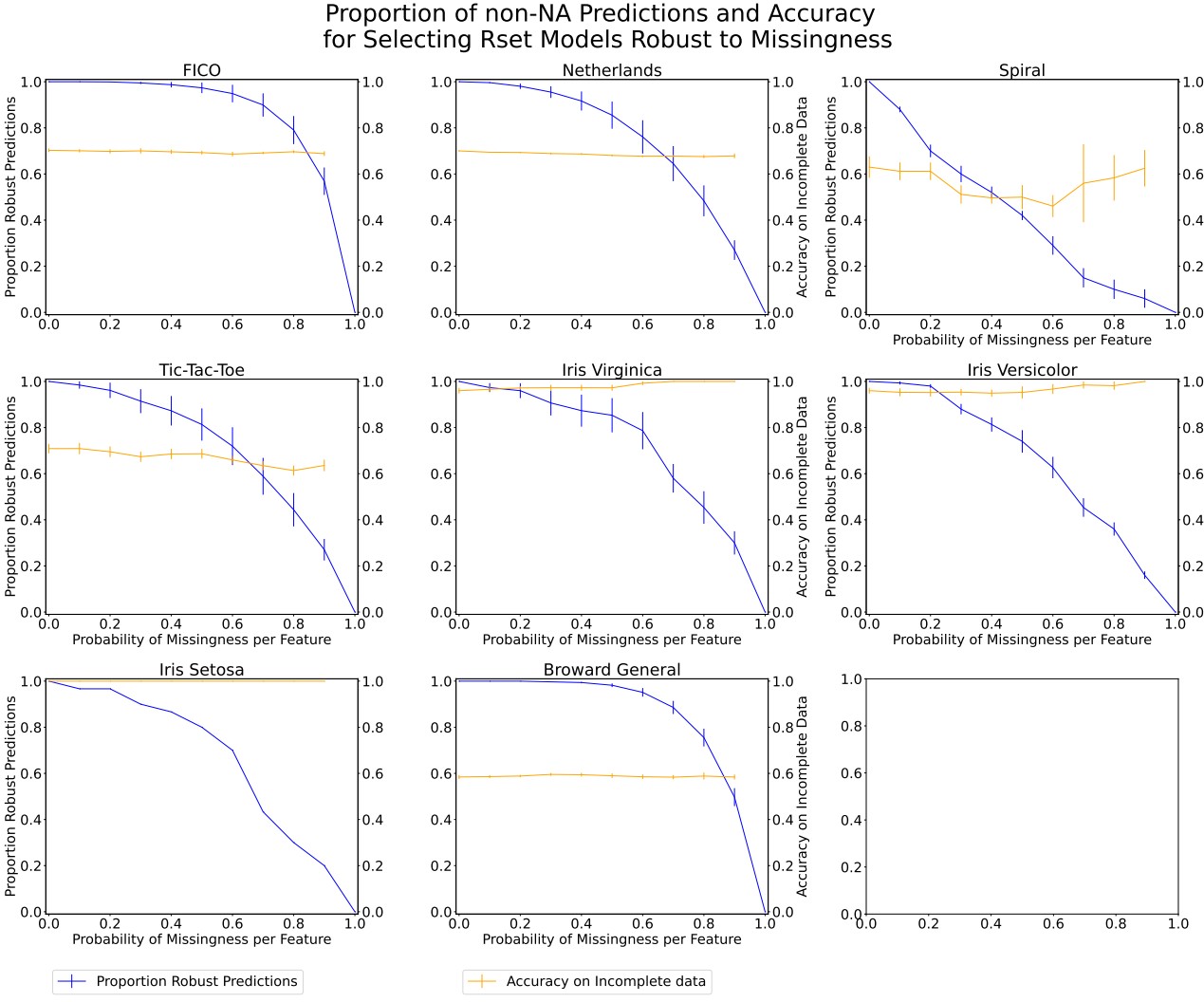

*Figure 12.* Rate at which at least one near-optimal tree can continue to make predictions as missing values are added, as well as accuracy on those predictions. These results use TreeFARMS (Xin et al., 2022) with maximum depth 3 and a standard per-leaf penalty of 0.01, finding all trees within 0.02 of the optimal objective.

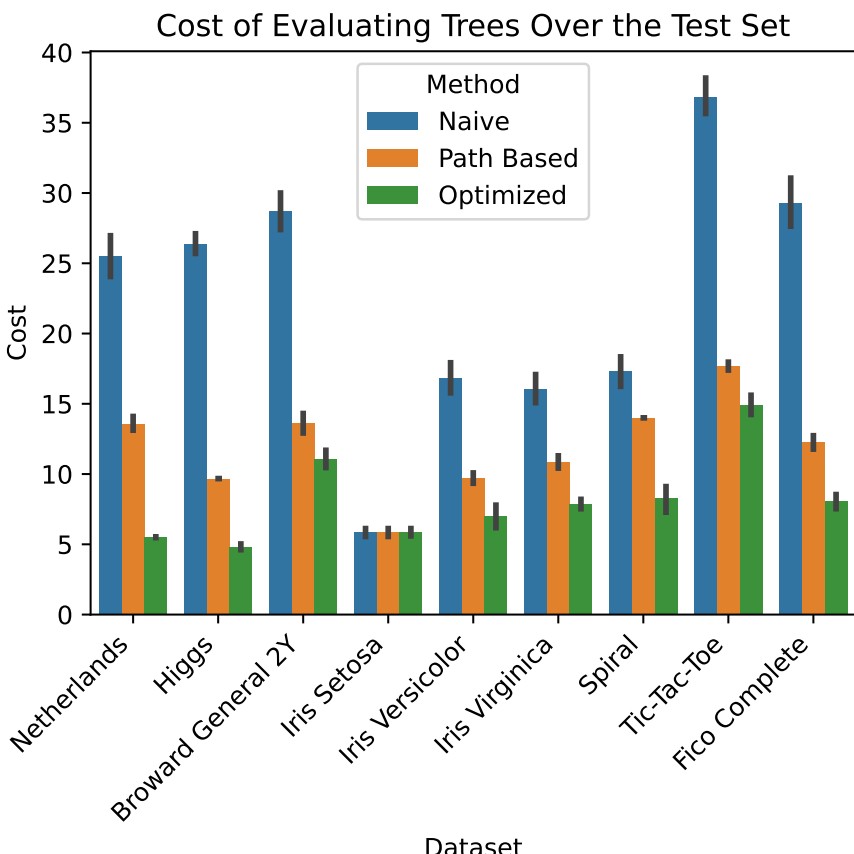

*Figure 13.* The cost of evaluating a tree by directly purchasing every variable in the tree (Naïve), purchasing variables in the order suggested by traversing the tree (Path Based), and by following our BCF/Q-learning policy (Optimized). Error bars report standard deviation of cost over 50 trees, each learned from a different bootstrap of the original dataset.

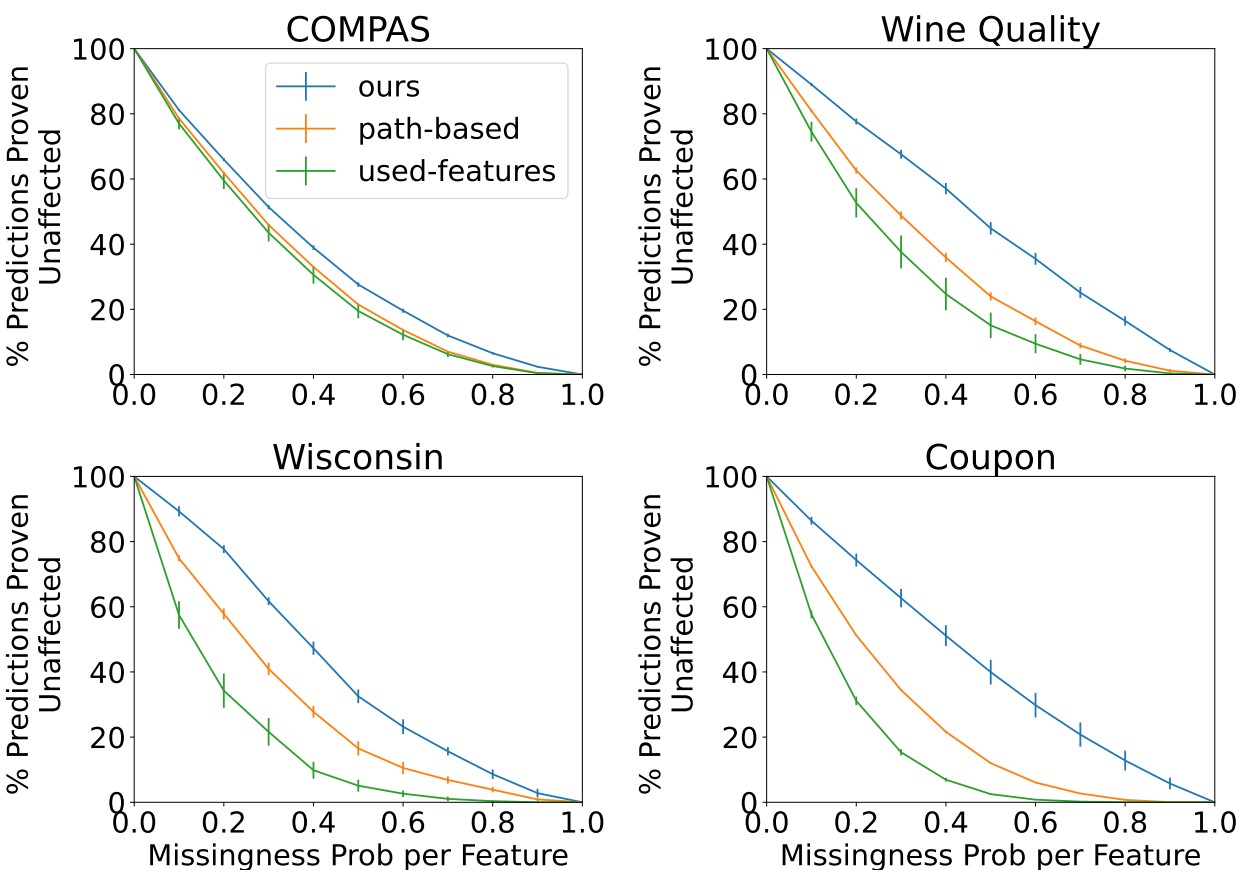

*Figure 14.* Rate at which decision trees can make predictions as missing values are added pre-binarization. These results use a simple CART tree, as implemented by SKLearn (Pedregosa et al., 2011), with depth 3 and default parameters.

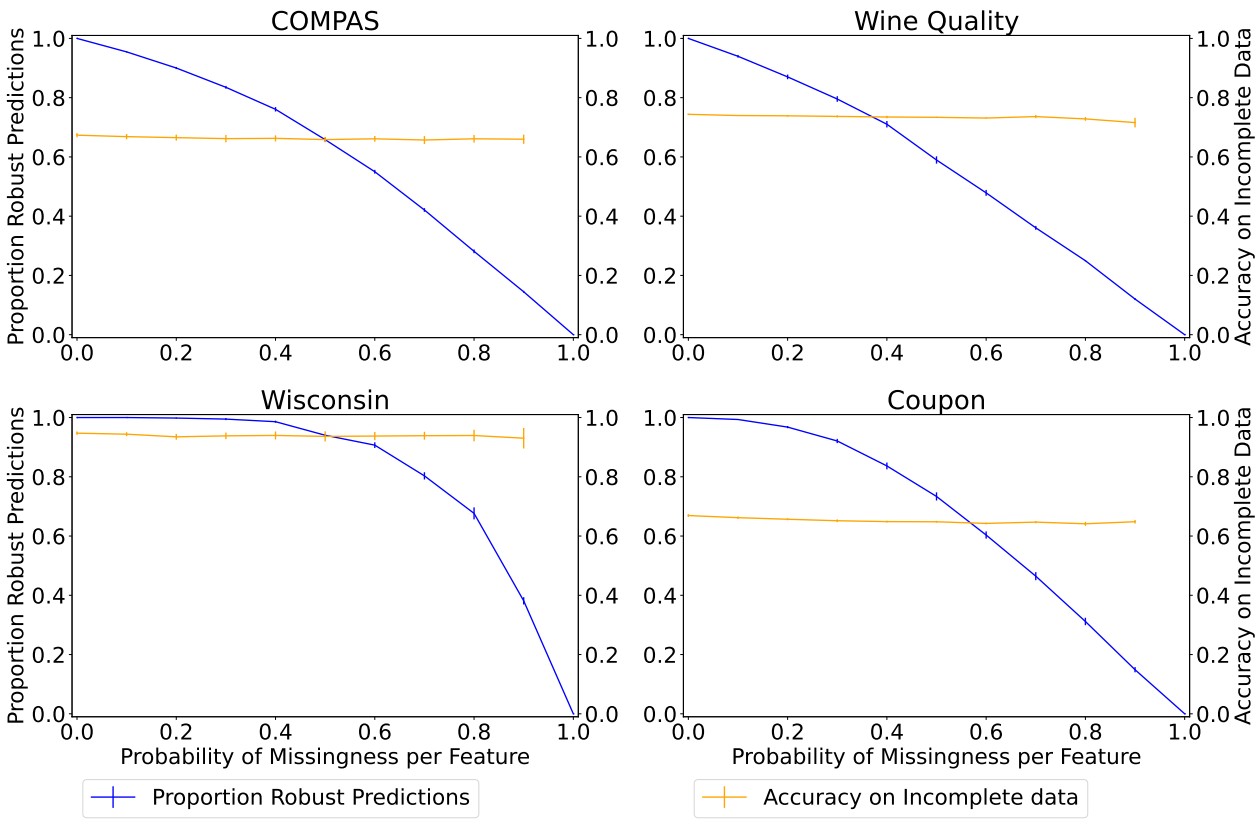

*Figure 15.* Rate at which at least one near-optimal tree can continue to make predictions as missing values are added pre-binarization. These results use TreeFARMS (Xin et al., 2022) with maximum depth 3 and a standard per-leaf penalty of 0.01, finding all trees with objective within 0.02 of optimal.

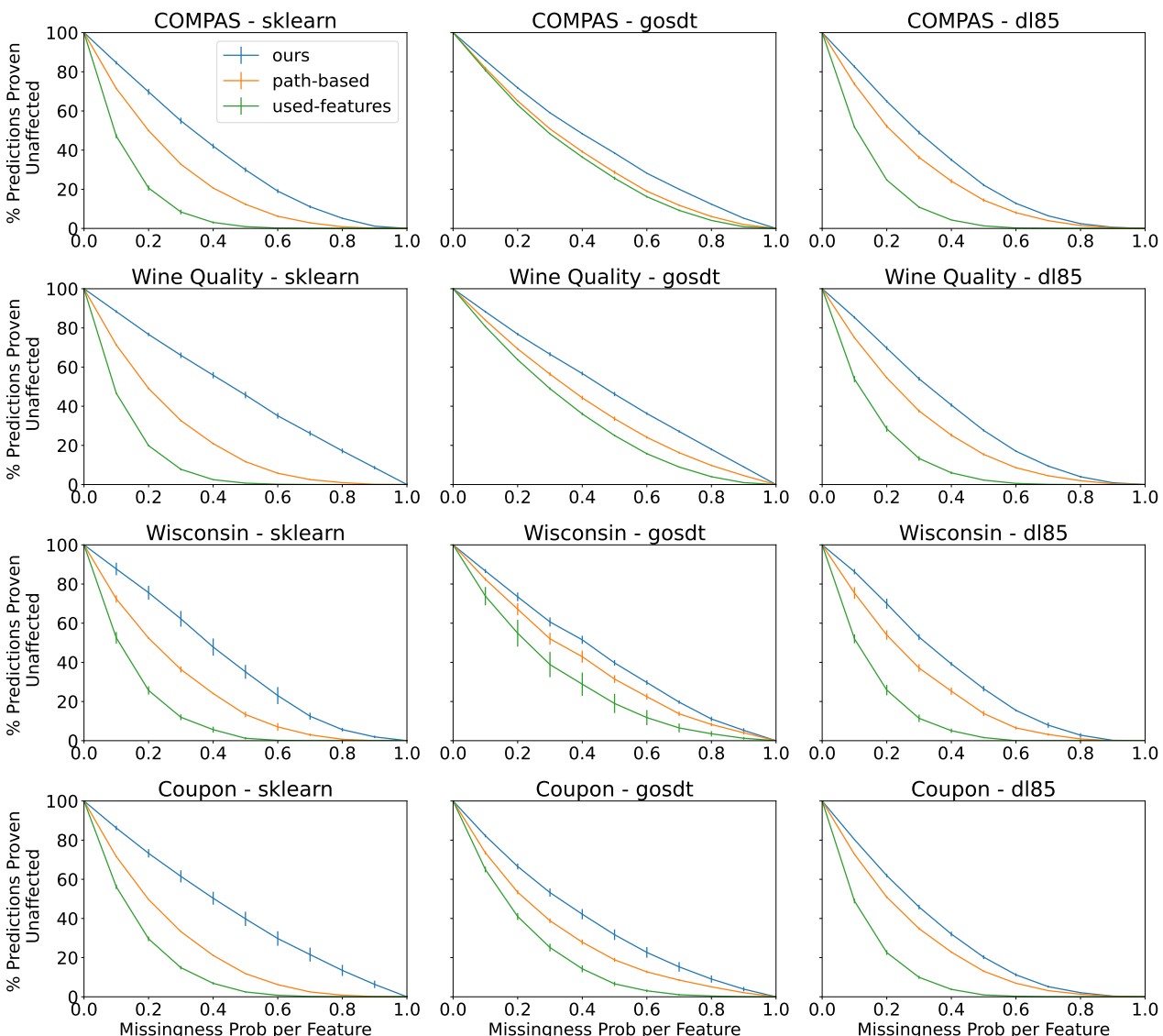

Figure 16. Rate at which decision trees can make predictions. These results use a simple CART tree, as implemented by SKLearn (Pedregosa et al., 2011), with depth 3 and default parameters, a simple GOSDT tree (Lin et al., 2020) with depth 3 and per-leaf penalty 0.01, and a simple dl85 tree (Aglin et al., 2020) with depth 3 and default parameters.

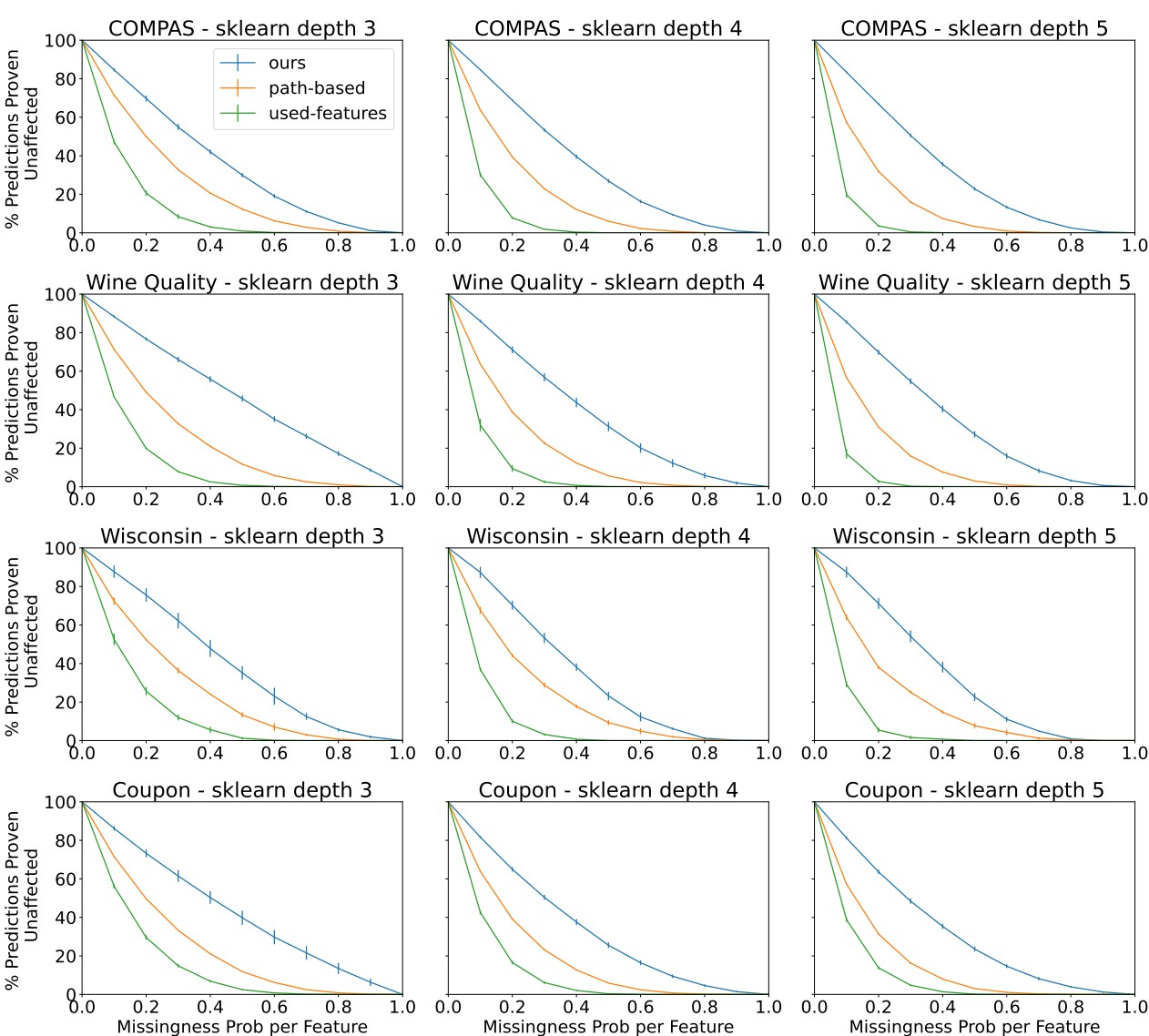

*Figure 17.* Rate at which decision trees can make predictions across decision tree depths. These results use a simple CART tree, as implemented by SKLearn (Pedregosa et al., 2011), with depth 3, 4, or 5 and default parameters.

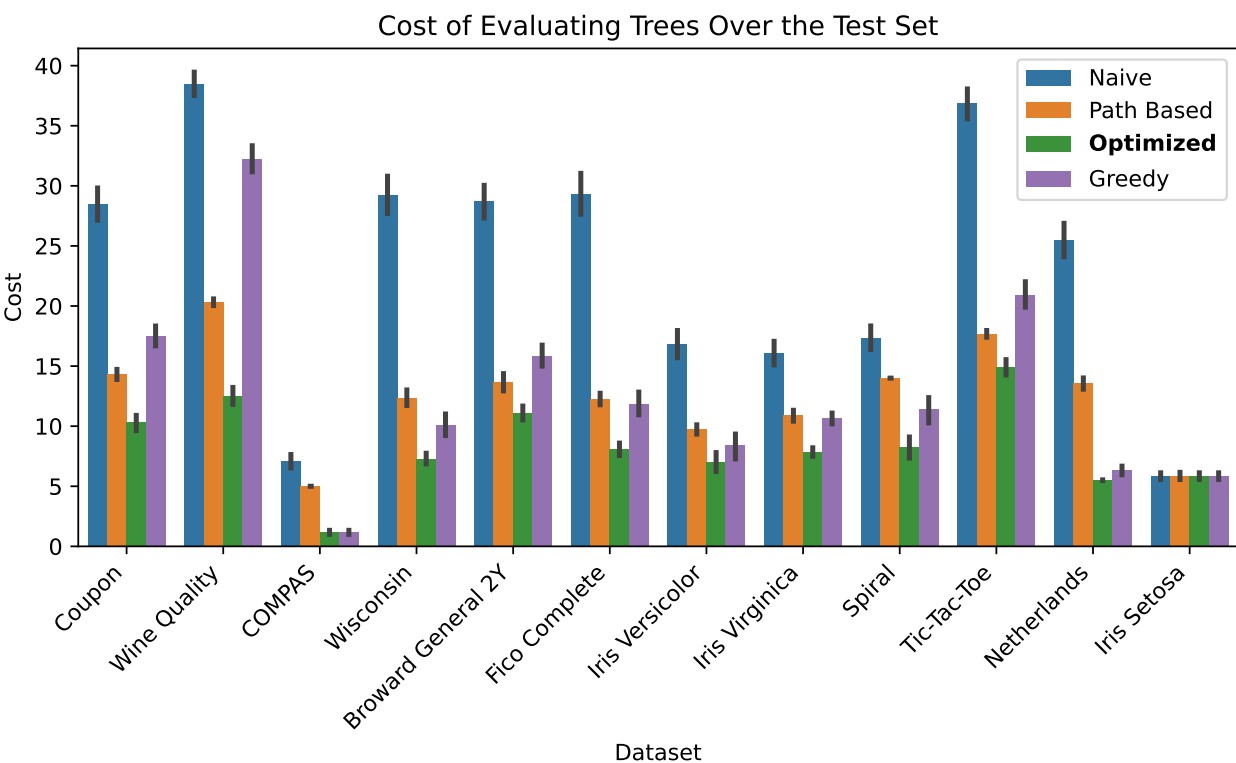

*Figure 18.* The cost of evaluating a tree by directly purchasing every feature in the tree (Naïve), purchasing features in the order suggested by traversing the tree (Path Based), following our BCF/Q-learning policy (Optimized), and by greedily purchasing the cheapest feature used by the tree until a clause in the BCF is satisfied (Greedy). Error bars report standard deviation of cost over 50 trees, each learned from a different bootstrap of the original dataset.

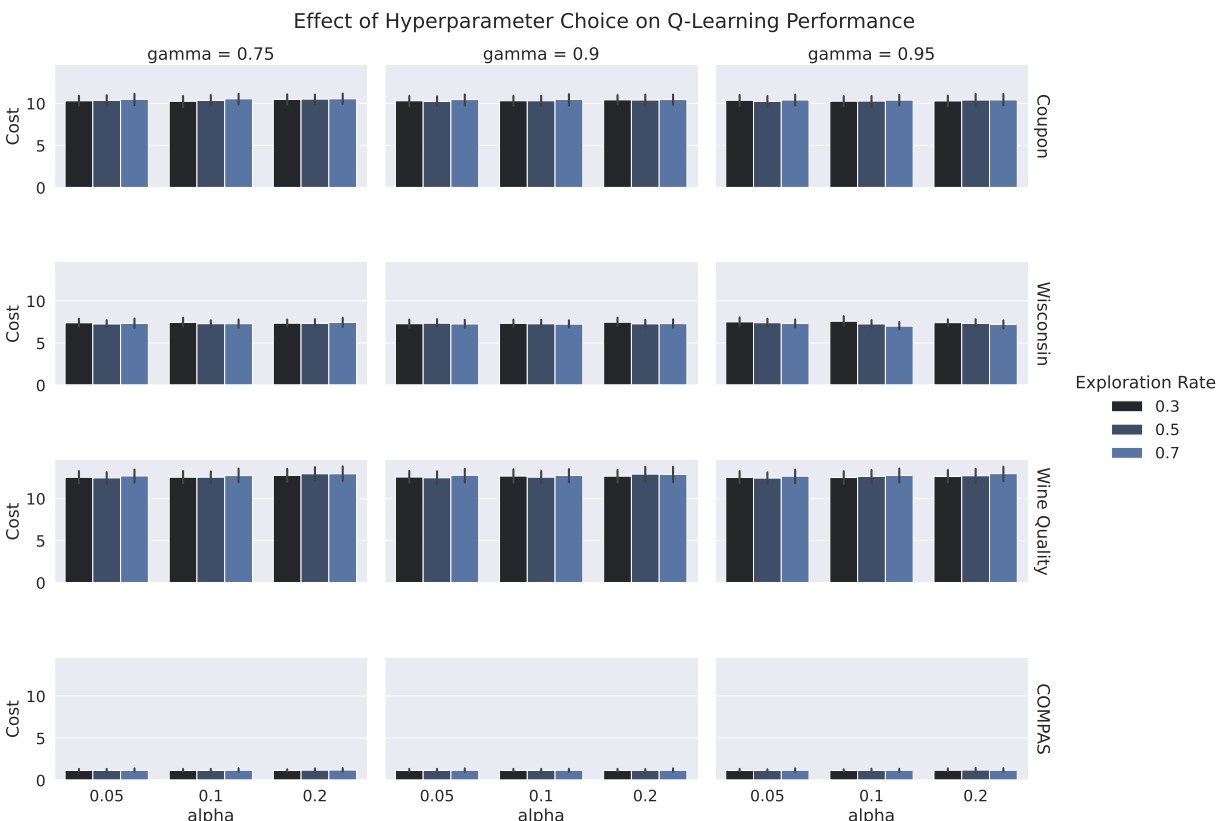

*Figure 19.* The cost of evaluating a tree by following our BCF/Q-learning policy under different Q-learning hyperparameters. Each row corresponds to a different dataset, each column a different value for gamma, each grouping along the x axis a different value of alpha, and each color a different exploration rate. Error bars report standard deviation of cost over 50 trees, each learned from a different bootstrap of the original dataset.

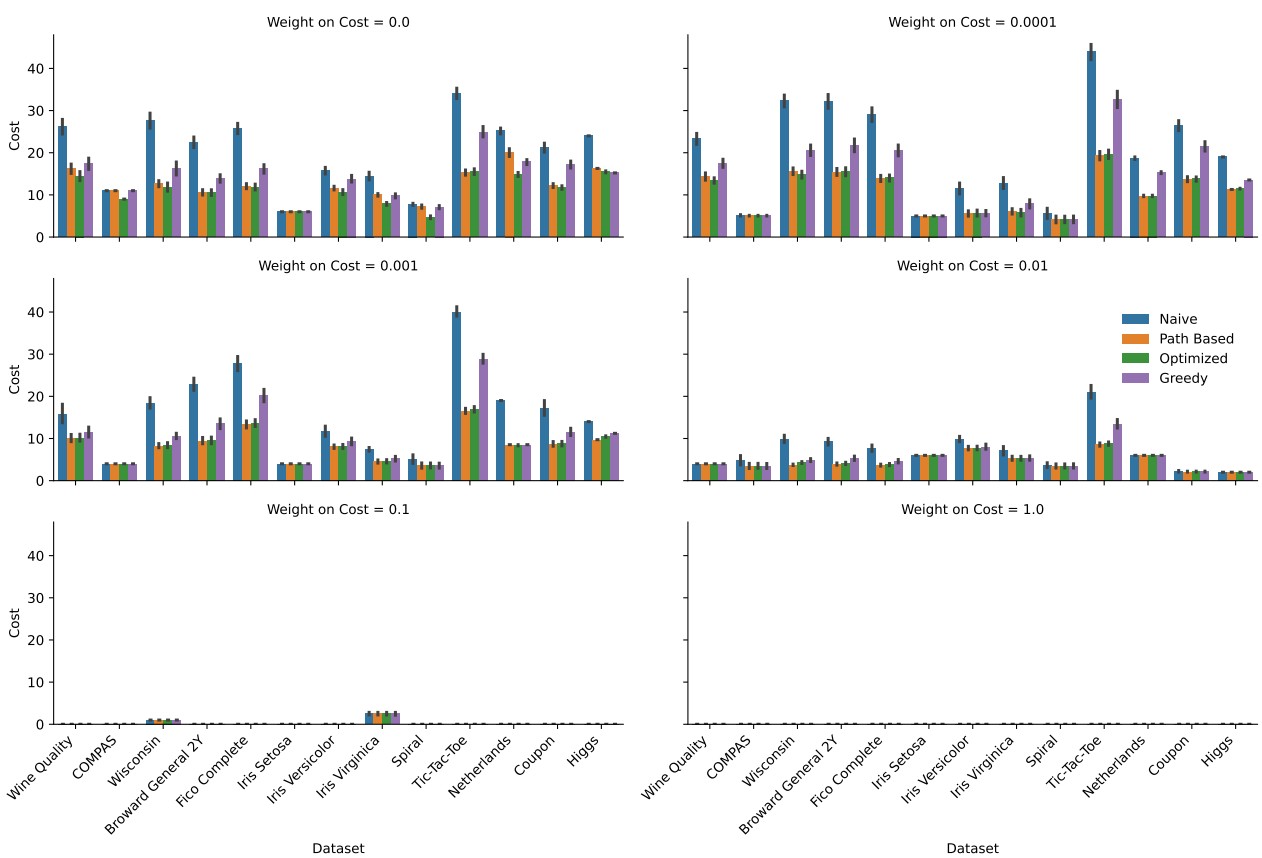

*Figure 20.* The cost of evaluating a cost-optimal tree under different cost weightings by directly purchasing every feature in the tree (Naïve), purchasing features in the order suggested by traversing the tree (Path Based), following our BCF/Q-learning policy (Optimized), and by greedily purchasing the cheapest feature used by the tree until a clause in the BCF is satisfied (Greedy). Error bars report standard deviation of cost over 50 trees, each learned from a different bootstrap of the original dataset.

