# OpenReview forum: "Leveraging Predictive Equivalence in Decision Trees"
_ICML.cc/2025/Conference — ICML 2025 poster_

### Official Review · Reviewer_1yAS · 2025-02-15

**Overall Recommendation:** 4

**Summary:**

The paper presents an intuitive boolean logical representation of decision trees. This representation removes predictive equivalence. They then show in 3 settings (feature importance, missing data, and improving cost efficiency) that this representation can yield improvements over standard tree representations.

**Claims And Evidence:**

Yes, the claims are clearly supported through the reduction of redundancies in the Rashomon set and the three case studies.

**Essential References Not Discussed:**

N/A

**Experimental Designs Or Analyses:**

Yes

**Methods And Evaluation Criteria:**

The authors propose interesting, non-obvious evaluations for their introduced representation.

**Other Comments Or Suggestions:**

N/A

**Other Strengths And Weaknesses:**

The paper is well written and the methods are clear. At first glance, the approach seemed potentially overly simplistic but I am convinced from the experiments that the author's contribution is substantial.

**Questions For Authors:**

Besides the Q-learning approach in Sec 7, is there a simpler optimization-based procedure for leveraging this representation to improve cost efficiency?

**Relation To Broader Scientific Literature:**

The authors do a good job contextualizing their paper in the Related works section. It fits into a large line of work that is displayed through their 3 case studies, and unites them in a simple way.

**Theoretical Claims:**

I briefly checked the proofs and believe they are correct.

---

> ### Author Rebuttal · Authors · 2025-03-31
>
> Thank you for your review! We are glad that you appreciated the novelty of our method through our experimental results.
>
> To answer your question, we are unaware of any substantially simpler procedure to improve cost efficiency. Naïvely, we could attempt to perfectly fit an optimal, cost sensitive decision tree to a version of the dataset in which the labels were replaced with predictions from our reference tree. However, to preserve predictive equivalence over all possible inputs, we would need to exactly fit to a version of the dataset that realized every possible set of input features, which could be prohibitively large (for binary data, $2^{\\# \\textrm{features}}$).
>
> We felt that the problem naturally lent itself to representation as a Markov decision process (MDP). While it is not viable to directly solve this MDP because of the extremely large state space, there are a number of alternative approaches to solving MDPs that could be considered, although we see none as notably simpler than Q-learning.
>
> Finally, if we were able to enumerate every predictively equivalent decision tree for a given DNF, we could simply iterate through them and select the most cost efficient one. However, we cannot yet map from a DNF to the set of equivalent trees. Understanding the group structure of predictively equivalent trees, as we briefly proposed as future work, would allow us to do this, but this remains to be solved. See our response to Reviewer Nmzb for more details on this direction.

---

### Official Review · Reviewer_AgS4 · 2025-03-14

**Overall Recommendation:** 4

**Summary:**

This paper addresses the issue of "predictive equivalence" in decision trees, where different tree structures can represent the same decision boundary but imply different evaluation procedures.  The authors propose a boolean logical representation using Disjunctive Normal Form (DNF) to abstract away the evaluation order and provide a representation faithful to the underlying decision boundary.  They demonstrate the utility of this representation in handling missing data, quantifying variable importance, and optimizing the cost of reaching predictions.

**Claims And Evidence:**

The claims in the submission are generally well-supported by clear and convincing evidence. The paper provides theoretical proofs in the appendix to support its claims.  Empirical evidence is provided through experiments on multiple datasets.

**Essential References Not Discussed:**

The paper adequately discusses relevant prior work. No essential references were identified that are not currently cited or discussed in the paper.

**Experimental Designs Or Analyses:**

The experimental setup is well-designed, with comparisons across multiple datasets and tasks. However, some limitations exist:

- The datasets used, while diverse, are relatively small. A study on larger datasets (e.g., industry-scale data) would be valuable.
- The method is compared primarily against baseline decision tree methods but lacks comparisons with other approaches that attempt to regularize or simplify decision trees.
- Sensitivity analysis on hyperparameters (e.g., Q-learning settings) is missing and could provide deeper insights into practical deployment.

**Methods And Evaluation Criteria:**

The proposed method of converting decision trees into DNF and using the Quine-McCluskey algorithm for simplification is appropriate for addressing the problem of predictive equivalence.  The evaluation criteria, including experiments on real-world datasets and comparisons with baseline methods, are sound and suitable for demonstrating the effectiveness of the proposed representation.

**Other Comments Or Suggestions:**

It would be beneficial to provide a more detailed discussion of the computational complexity of Algorithm 1 and the Quine-McCluskey algorithm, especially in relation to the size and depth of the input decision tree.

**Other Strengths And Weaknesses:**

Strengths:
* The paper is well-written and clearly explains the problem of predictive equivalence in decision trees.
* The proposed DNF representation is a novel and effective approach to address this problem.
* The paper provides strong theoretical and empirical evidence to support its claims.
* Applications of the proposed representation to missing data, variable importance, and cost optimization are relevant and well-demonstrated.

Weaknesses:
* The computational feasibility of the approach for deep trees is not discussed in depth.
* Comparisons with alternative tree simplification methods are missing.

**Questions For Authors:**

* How does the computational complexity of the DNF transformation scale with tree depth and dataset size?
* Could the proposed representation be extended to ensemble methods like random forests?
* How does the method compare to approaches using Binary Decision Diagrams (BDDs)?

**Relation To Broader Scientific Literature:**

The key contributions of the paper are well-related to the broader scientific literature. The authors discuss prior work on decision tree learning, variable importance, missing data, and cost optimization.  They clearly articulate how their work builds upon and extends previous research in the field.

**Theoretical Claims:**

The paper includes several theoretical claims, such as the faithfulness of the DNF simplified form, completeness, succinctness, and resolution of predictive equivalence.  The correctness of the proofs for these claims was checked, and no issues were identified.

---

> ### Author Rebuttal · Authors · 2025-03-31
>
> Thank you for your thoughtful and thorough review! We hope that the comments below address your concerns.
>
> **Complexity of Algorithm 1 and Quine-McCluskey**
>
> The DNF simplification problem solved by the Quine-McCluskey algorithm is NP-Complete in the number of variables used by the tree. Since trees never split on the same binary variable twice in a path from root to leaf, our problem is NP-Hard in tree depth (a tree of depth $d$ can split on between $d$ and $2^d-1$ features). The DNF transformation is applied to trees that have already been fit, so the complexity of the transformation depends only on the size of the tree, not the size of the dataset.
>
> We do not mind this worst-case runtime (DNF simplifications were very fast in our experiments), because decision trees are often favored in practical problems that are noisy and for which data is not abundant. In this scenario, prior work has shown that simpler decision trees are near optimal (see the end of Section 2.1 in our paper), so in this situation we are solving small instances of the DNF simplification problem. If accepted, we will use the extra page to further elaborate on the computational complexity of our algorithms as it relates to the practical problems we seek to address.
>
> **Alternative Tree Simplification Methods**
>
> It is important to note that our method applies to decision trees found via any algorithm, even those which inherently regularize or otherwise simplify trees. Izza et al. ([1]) enumerated many decision trees from textbooks and research papers exhibiting explanation redundancy, and thus predictive equivalence (see Proposition 3.3 in our paper). Some of these trees were found with methods that regularize or simplify trees as part of an optimization objective, yet they still exhibited predictive equivalence.
> In our experiments with the Rashomon set of sparsity-regularized decision trees, we found many predictively equivalent models, even though all models in the Rashomon set were quite simple.
> Thus, the conclusions of our experiments likely wouldn't be affected by changing the method of decision tree optimization.
>
> While alternative post-processing approaches could be applied to simplify a given decision tree (e.g., specialized pruning), we are unaware of any such approach that is guaranteed to produce a predictively equivalent tree. As such, these methods are not solving the same problem as ours.
>
> **Random Forests**
>
> Yes, the proposed representation could absolutely be extended to ensemble methods such as random forests! See our response to Reviewer Nmzb for details.
>
> **Binary Decision Diagrams**
>
> While exploring methods to resolve predictive equivalence, we did not see a particular benefit to using binary decision diagrams over our chosen DNF representation. It is NP-Hard to find the variable ordering leading to the simplest BDD, which we would need to find in order to fully resolve predictive equivalence. Thus, BDDs do not offer us a direct computational advantage over the Quine McCluskey algorithm. We also feel that it is easier to interpret a DNF's terms as 'reasons' to predictive positive than it is to understand a tree's predictive behavior from a BDD.
>
> **Dataset Size**
>
> We've extended the results in figures 6 and 8 to a one million sample version of the Higgs dataset (https://www.openml.org/search?type=data&sort=runs&id=42769&status=active). The results are similar to those of other datasets in our paper and can be found here: https://docs.google.com/document/d/e/2PACX-1vR-i5kxlIeEK1tBIBFqXOcxhHaXqs6WQPTmjzRv7iIaNy90zevAiX8YawK2ICib0cu-tX6uG9SQdqCM/pub.
>
> We would also like to note that two of the further datasets in our appendix -- Netherlands and FICO -- each have more than 10,000 training samples and are from real-world industry (FICO) or government (Netherlands) settings.
>
> **Hyperparameter Sensitivity of Q-learning**
>
> Thank you for this suggestion. We performed a sweep over reasonable values for the key hyperparameters in the Q-learning framework, the results of which can be seen by following https://docs.google.com/document/d/e/2PACX-1vRCgUtuqu7AX9sOjM6AOXzj0ieyNe-krvzoj9048Wb-SkbH6-n3dZQOIQ3bPksGgRelMC77gqY96MSO/pub. We find that our Q-learning approach yields similar evaluation costs to the values reported in the main paper across 27 different hyperparameter settings.
>
> [1] Izza, Y., Ignatiev, A., and Marques-Silva, J. On tackling
> explanation redundancy in decision trees. Journal of
> Artificial Intelligence Research, 75:261–321, 2022.

---

### Official Review · Reviewer_fcd7 · 2025-03-16

**Overall Recommendation:** 3

**Summary:**

The authors propose the use of a minimal Boolean formula as a DNF in order to represent a decision tree that has been learnt.  This representation is useful for the authors, as one no longer requires the evaluation of the learnt function to happen in a top-down manner (i.e., start from the root node of the tree and proceed all the way down to some leaf).  A consequence that is investigated in the paper, is how this new representation can help with missing attribute values.  Another point of investigation is in terms of explainability, where it is shown that different trees that are predictively equivalent may indicate very different attributes as being the most important ones.  Finally, there can also be applications of the DNF representation to cost-sensitive applications.  The authors provide some theoretical results early on that justify the faithfulness, completeness, succinctness of the proposed DNF form with respect to some decision tree, as well as a theorem explaining that structurally different decision trees that are nevertheless predictively equivalent to will have the same minimal DNF form.  Near the end of the paper, the effectiveness of the proposed method is investigated using four different data sets: COMPAS, wine quality, Wisconsin, and Coupon.

**Claims And Evidence:**

Yes, to the extent that I checked, the claims are adequately supported by convincing evidence; be it theorems or experiments.

**Essential References Not Discussed:**

I cannot think of something.  I think the authors have done a good job citing existing literature.

**Experimental Designs Or Analyses:**

I did go through the experiments as these are described in the paper.  The results seem very intuitive to me and along the lines of what I would expect to see.

**Methods And Evaluation Criteria:**

Yes, the proposed methods and evaluation criteria make sense.  They are actually pretty standard.

**Other Comments Or Suggestions:**

I think the authors have done a good job writing a good paper.  Nevertheless, some part of the paper can be improved.

1. The Rashomon set is never defined.  It should be defined somewhere in the text.

2. Along these lines, I believe TreeFARMS is being used in order for the authors to be able to argue about the Rashomon set.  Hence, I would personally prefer to see a small paragraph describing the mechanics of TreeFARMS, so that it is easy for everyone to understand what is happening without too much effort.

3. Lines 258-259: X_0, X_1 -> X_1, X_2

4. Figure 3: Please describe the information that what we see in each node.

**Other Strengths And Weaknesses:**

As mentioned above, I do not believe that this paper is offering any new theoretical insights.  However, the experimental part does provide substantial new information (to the extent that I know) and the insights on missing attribute values, the importance of various attributes to be more robust, and the use of the ideas in situations where we have different costs in order to obtain attribute values, are all very important in the real world and therefore the paper has a lot of merit.  Furthermore, the presence of known theoretical results allows someone to appreciate better the content of the paper (I am one of them).

**Questions For Authors:**

Q1.  One of the issues that is presented near the beginning of the paper explains how two structurally different trees that are equivalent w.r.t. the function that they compute may give rise to situations where one of them is able to process an instance with missing values, while the other one may not.  This is due to the order by which one evaluates attributes from root to a leaf.  What is not clear to me is how a DNF formula that is equivalent with both of the above trees would not have such an issue.  Can you please clarify?

**Relation To Broader Scientific Literature:**

I appreciate the insights that are gained for situations where we have missing data as well as the insights in terms of which attributes are important for explainability concerns.

**Theoretical Claims:**

The proofs of the theorems are in the appendix.  In all honesty I did not thoroughly check the proofs.  The reason is that all these results should be known for perhaps 40+ years now (i.e., since the beginning of decision-tree theory.  The authors are basically saying that any decision tree can be represented in DNF form and argue about this in three different theorems (faithfulness, completeness, succinctness).  For example, this is discussed in a book from the 90's (Machine Learning, by Tom Mitchell).  As for the other result on having a unique minimal description of a DNF that corresponds to two predictively equivalent trees; well, this should also be known for 40+ years in the context of Boolean functions.  I appreciate the exposition of these results in the paper, as well as the fact that the proofs are stated in the appendix, but there is no surprise here and most likely no new result that has not been known for four decades now.  Nevertheless, again, there is merit that these results are part of the paper, and they should stay.

---

> ### Author Rebuttal · Authors · 2025-03-31
>
> Thank you for the thorough and thoughtful review and for the suggested additions in describing the
> Rashomon set and TreeFARMS. If accepted, we will use our extra page to incorporate this discussion into
> the camera-ready version. Thank you also for flagging the off-by-one typo in lines 258-259, and the need to
> clarify figure 3.
>
> **Q1. Evaluating a DNF with Missing Values**
>
> When we have a sample with missing values, we substitute all known values from that sample into our DNF formula. If a term in either the positive or negative DNF is satisfied, we return 1 or 0, resp. If not, then we simplify the expression again and check if it simplifies to 1 or 0. For the example in question, if $X_1$ is unknown and $X_2 = 0$ (or vice versa), then one of the forms of the tree will be unable to predict with the usual path-based evaluation method. However, substitution into the DNF expression gives $X_1 \land X_2 \to X_1 \land 0 \to 0$. Alternatively, if $X_1$ is unknown, and $X_2 = 1$, then substitution yields $X_1 \land 1 = X_1$, and we know that no predictively equivalent form of the tree will be able to make a prediction without knowing $X_1$.
> It is easy to see this in our toy example, but in more complicated trees this phenomenon can be much harder to identify without our approach.

---

### Official Review · Reviewer_Nmzb · 2025-03-17

**Overall Recommendation:** 3

**Summary:**

The paper addresses the challenge of predictive equivalence in decision trees, where multiple trees with identical decision boundaries but different evaluation processes complicate model selection. To resolve this, the authors propose a Boolean logical representation of decision trees that eliminates predictive equivalence, ensuring faithfulness to the underlying decision boundary, and demonstrate its applications to robustness under missing feature values, variable importance quantification, and prediction cost optimization.

**Claims And Evidence:**

Yes

**Essential References Not Discussed:**

N/A

**Experimental Designs Or Analyses:**

I have reviewed the first two case studies, and I believe they are valid.

**Methods And Evaluation Criteria:**

Yes

**Other Comments Or Suggestions:**

N/A

**Other Strengths And Weaknesses:**

Strength: The theoretical and experimental work in this paper is comprehensive, the motivation is clear, and it meets the theoretical standards expected of an ICML paper.

Weaknesses：
1.The research problem addressed in the paper is somewhat outdated.

2. I would like to ask how the method proposed in this work and the methods discussed in the future outlook, such as the group structure of decision trees and forest methods like random forests, differ from each other.
f the author provides a good answer to this question, I would be willing to raise my score.

**Questions For Authors:**

Please refer to the secion of the Weaknesses.

**Relation To Broader Scientific Literature:**

N/A

**Theoretical Claims:**

I have reviewed all the theorems in Section 3.1, and I believe they are both profound and meaningful.

---

> ### Author Rebuttal · Authors · 2025-03-31
>
> Thank you for the review! We are glad that you appreciated our theory, experiments, and motivation. We would be happy to engage in further discussion on the modernity of our work. We interpreted Weakness 1 as saying that research on decision trees is outdated -- if we misunderstood the weakness, please correct us.
>
> While decision trees have long been established as a type of machine learning model, their study is still relevant. Recent advances in both theory (see [1] for the existence of competitive simple models) and practice (see [2] for advances in decision tree optimization) have shown that, for many datasets, a single decision tree can achieve state-of-the-art performance. In addition, thanks to recent progress in both computing and in algorithms research, decision trees are now one of the few model classes for which study of the entire Rashomon set is possible [3].
>
> The problem of predictive equivalence is relevant in modern decision tree literature. Predictive equivalence occurs in any model that is built upon decision trees, including random forests and boosting models, meaning the findings in this paper apply to a wide class of models.  In our response to reviewer AgS4, we discuss a recent paper that found that modern tree optimization algorithms often construct trees with redundant path explanations, which we show are related to predictive equivalence. Predictive equivalence is also relevant for modern Rashomon set research - see section 4, where we show predictive equivalence is rampant throughout the Rashomon set, and section 5.2, where we explore the implications on a variable importance task that leverages the Rashomon set.
>
> **The Group Structure of Decision Trees**
>
> Our approach defines a natural equivalence relation, characterized by identical logical formulae, for decision trees. This relation identifies when two or more trees are predictively equivalent, but we know of no efficient way to enumerate every predictively equivalent form of a given tree. If we knew the operations that could be performed on a tree to generate all the other trees in its equivalence class, we could materialize all (non-trivial) predictively equivalent trees to a particular tree. As a byproduct, this would improve the efficiency of our cost optimization approach (see our response to Reviewer 1yAS).
>
> **Random Forests**
>
> Our approach focuses on the ramifications of predictive equivalence for individual trees. Future work could explore the consequences of predictive equivalence on ensembles. The simplest approaches for this might involve exploring predictive equivalence for each component tree, and investigating how costs, robustness to missing data, and variable importance change across different versions of the same ensemble, where individual trees are replaced with predictively equivalent alternatives. There's also a possibility to explore the random forest as a single large decision tree (see, for example, [4]) to identify additional predictive equivalence beyond what can be observed by equivalence of individual component trees. These single trees can contain hundreds of nodes, however, so analyzing the entire tree all at once would require innovations in scalability as a part of future work.
>
> [1] Semenova, L., Rudin, C., and Parr, R. On the existence of
> simpler machine learning models. In 2022 ACM Conference on Fairness, Accountability, and Transparency, pp.
> 1827–1858, 2022.
>
> [2] Costa, V.G., Pedreira, C.E. Recent advances in decision trees: an updated survey. Artif Intell Rev 56, 4765–4800 (2023). https://doi.org/10.1007/s10462-022-10275-5
>
> [3] Xin, R., Zhong, C., Chen, Z., Takagi, T., Seltzer, M., \& Rudin, C. (2022). Exploring the whole rashomon set of sparse decision trees. Advances in neural information processing systems, 35, 14071-14084.
>
> [4] Vidal, T., Schiffer, M. Born-again tree ensembles. International Conference on Machine Learning. PMLR, 2020. https://arxiv.org/abs/2003.11132

---

### Decision · Program_Chairs · 2025-05-01

**Decision:**

Accept (poster)

**Comment:**

This paper addresses the issue of predictive equivalence in decision trees, i.e., the existence of decision trees that represent the exact same decision boundary but differ in their evaluation processes. The authors present a boolean logical representation of decision tree that does not exhibit predictive equivalence. The positive impact of this representation is highlighted in three downstream tasks: missing value handling, variable importance measures and cost efficiency optimization.

The reviewers have praised the quality of the writing, the novelty and effectiveness of the proposed representation, the soundness of the theoretical analyses, the diversity and relevance of the experiments. They have also identified a few weaknesses, such as the limited size of the datasets used in the experiments, some lack of novelty in the proofs, and the high computational cost of the approach. The authors have, however, convincingly addressed them in the rebuttal, for example, by providing new results on larger datasets. The incorporation of these discussions and new results will improve the paper's overall quality.

In light of the reviewers’ recommendations, I suggest accepting the paper. It represents a solid contribution to the field.